

# Sensitivity of GPS tropospheric estimates to mesoscale convective systems in West Africa

Samuel Nahmani[1], Olivier Bock[1], Françoise Guichard[2]

[1]IPGP, IGN, Université Paris Diderot, Sorbonne Paris Cité, UMR 7154 CNRS, Paris, France
[2]CNRM, CNRS UMR 3589 and Météo-France, Toulouse, 31057 Cedex 1, France

*Correspondence to*: Samuel Nahmani (Samuel.Nahmani@ign.fr)

**Abstract.** This study analyzes the characteristics of GPS tropospheric estimates (Zenith Wet Delays, and gradients, and post-fit phase residuals) during the passage of Mesoscale Convective Systems (MCSs) and evaluates their sensitivity to the research-level GPS data processing strategy implemented. Here, we focus on MCS events observed during the monsoon seasons of West Africa. This region is particularly well suited because of the high frequency of occurrence of MCSs in contrasting climatic environments between the Guinean coast and the Sahel. This contrast is well sampled data with the six AMMA GPS stations. Tropospheric estimates for 3-year period (2006-2008), processed with both GAMIT and GIPSY-OASIS software packages, were analyzed and inter-compared. First, the case an MCS which passed over Niamey, Niger, on 11 August 2006, demonstrates a strong impact of the MCS on GPS estimates and post-fit residuals when the GPS signals propagate through convective cells as detected on reflectivity maps from MIT's C-band Doppler radar. The estimates are also capable of detecting changes in the structure and dynamics of the MCS. The sensitivity is however different depending on the tropospheric modeling approach adopted in the software. With GIPSY-OASIS, the high temporal sampling (5 min) of Zenith Wet Delays and gradients is well suited for detecting the small-scale, short-lived, convective cells, while the post-fit residuals remain quite small. With GAMIT, the lower temporal sampling of the estimated parameters (hourly for Zenith Wet Delays and daily for gradients) is not sufficient to capture the rapid delay variations associated with the passage of the MCS, but the post-fit phase residuals clearly reflect the presence of a strong refractivity anomaly. The results are generalized with a composite analysis of 414 MCS events observed over the 3-year period at the six GPS stations with the GIPSY-OASIS estimates. A systematic peak is found in the Zenith Wet Delays coincident with the cold-pool crossing time associated to the MCSs. The tropospheric gradients are reflecting the path of the MCS propagation (generally from East to West). This study concludes that Zenith Wet Delays, gradients, and post-fit phase residuals provide relevant and complementary information on MCSs passing over or in the vicinity of a GPS station.

## 1    Introduction

The American Meteorological Society (2015) defines a Mesoscale Convective System (MCS) as "a cloud system that occurs in connection with an ensemble of thunderstorms and produces a contiguous precipitation area on the order of 100 km or more in horizontal scale in at least one direction. An MCS exhibits deep, moist convective overturning contiguous with or embedded within a mesoscale vertical circulation that is at least partially driven by the convective overturning". Figure 1 pictures a typical night-time situation of the convective activity over





West Africa during the monsoon season, with two active clusters of MCS on eastern and western borders of Nigeria, and more scattered dissipating MCSs over Guinea. Small isolated developing MCSs can also be seen in northern Ghana and over the Senegal-Mauritania border. The MCS's internal structure typically comprises convective and stratiform regions as described by Zipser et al. (1977), Houze et al. (1990) or Houze (1989; 1997;

2004). However, numerous questions related to MCSs remain only very partly answered, such as the mechanisms behind their life cycle (initialization, propagation, decay, e.g. Laing et al., 2008), and how these mechanisms involve interactions between physical processes (e.g. surface and boundary layer processes - Taylor et al., 2017; Rochetin et al., 2017) and atmospheric circulations (e.g. African easterly waves - Fink et al., 2006; Rickenbach et al., 2009; Beucher et al., 2014).

Deep convection, MCSs and rainfall are strongly linked to atmospheric water vapor (e.g. Bretherton et al., 2004; Schiro et al., 2016; Sherwood et al., 2010). This is why GPS integrated water vapor (IWV) data have been used for studies in various tropical climates such as the India monsoon (Jade et al., 2005; Jade and Vijayan, 2008), the South and North American monsoon (Gutzler et al., 2004; Means, 2013; Adams et al., 2011 and 2014; Serra et al., 2016) or the West African monsoon (WAM) (Couvreux et al., 2011; Barthe et al., 2010). Here, we

focus on MCSs occurring during the WAM, which has been the subject of intense investigations in recent years in the framework of the AMMA program (Redelsperger et al., 2006; Lafore et al., 2011). Indeed, MCSs are a key component of the WAM, as they produce 90% of annual rainfall in West Africa (D'Amato and Lebel, 1998). Within AMMA's framework, six GPS stations (Fig. 1) were operated between 2005 and 2010 to sample integrated water vapor (IWV) along the climatic gradient from the arid saharo-sahelian climate in Mali to the

moist sudano-guinean climate in Benin (Bock et al., 2008). The AMMA GPS data have proved to be valuable for understanding the water cycle of this region: WAM atmospheric processes (Janicot et al., 2008 ; Lafore et al., 2011), land water storage changes (Hinderer et al., 2009), water budgets (Meynadier et al., 2010a; 2010b) or hydrological loading deformation induced by WAM (Nahmani et al., 2012).

    Links between atmospheric water vapor and convective precipitation occur on various scales down to less

than a few hours. Notably, GPS data have revealed that a sharp IWV increase is often observed with the passage of MCSs (Barthe et al., 2010; Schiro et al., 2016; Taylor et al., 2017). In this study, we focus on this 'small' scale (a few hours or less). More precisely, we address the question of the accuracy of AMMA GPS observations and the information content of tropospheric delay estimates during the passage of MCSs over West Africa comparing the two main research-level GPS data processing approaches: network processing in which

differentiated or undifferentiated observations from several stations are processed simultaneously and Precise Point Positioning (PPP) in which each station is processed independently but requires precise satellite orbits and clocks as calculated by the International GNSS Service (IGS) or one of its analysis centers (www.igs.org). In both approaches, many processing options can be used, depending on the functionality of the software: least squares adjustment or Kalman filter, choice of the elevation cutoff angle, modeling of the troposphere, rejection

procedure of the erroneous observations (screening), etc. In this paper we compare the GPS Analysis at MIT (Massachusetts Institute of Technology) (GAMIT) software (Herring et al., 2015) and the GNSS Inferred Positioning System and Orbit Analysis Simulation software (GIPSY-OASIS) from the NASA Jet Propulsion Laboratory/Caltech (https://gipsy-oasis.jpl.nasa.gov/). The first performs a least squares adjustment on a regional network with one zenith tropospheric delay (ZTD) estimated every hour while the second uses a Kalman filter to

process the data in PPP mode with one ZTD estimated every 5 minutes. Section 2 discussed the processing



options available in both software and the expected impact on the tropospheric delay estimates. Results are presented and discussed for the AMMA GPS stations over a continuous period of 3 years (2006 to 2008, the most complete period for the six stations). The climatic context of West Africa lends itself particularly well to this methodological study with marked dry and wet seasons. Section 3 presents the case study of an organized cluster of MCSs passing over Niamey, Niger, on 11 August 2006 (Fig. 1). It illustrates the main characteristics of MCS events captured by GPS tropospheric estimates according to the tropospheric modeling used during the data processing. Section 4 provides at statistical characterization of MCSs using GPS tropospheric delay estimates from 414 events observed by the six AMMA stations between 2006 and 2008. Section 5 concludes the study.

## 2    GPS data processing and methods

The six AMMA GPS stations used here meet the quality standards of the IGS (Dow et al., 2009) and are moreover equipped with meteorological sensors (Vaisala, PTU200). Their Receiver-Independent Exchange (RINEX) files are freely available on IGS servers (see section data availability).

### 2.1    GPS Data Analysis with GAMIT

The RINEX data from the six AMMA GPS stations are processed from 1 January 2006 to 31 December 2008 within a regional network of around 25 IGS stations with the GAMIT scientific software (Herring et al., 2015) release 10.6. The settings of the GPS data processing are designed to meet the recommendations of the IERS conventions (Petit and Luzum, 2010) and correspond to an update to those used by Bock et al. (2008). The IGS final orbits are held fixed. A priori station positions expressed in ITRF2014 (Altamimi et al., 2016) are constrained with 10 cm and 30 cm a priori standard deviations for horizontal and vertical components respectively. GPS phase observations are corrected applying absolute antenna phase calibration models (Schmid et al., 2007) and weighted using the elevation-dependent variance function $\sigma^2 = a^2 + b^2/\sin^2(e)$, where $e$ is the elevation angle and $a$ and $b$ are determined by a least squares of the post-fit residuals in an iterative procedure. The integer ambiguities are determined using wade-lane and narrow-lane combinations (Dong and Bock, 1989). The elevation cutoff angle is fixed to 7°. Ocean tide loading effects are corrected using the FES2004 model (Lyard et al., 2006). Atmospheric loading and non-tidal ocean loading can be neglected at AMMA GPS stations (Nahmani et al., 2012). This approximation introduces an uncertainty at a submillimetric level on ZWD estimates at AMMA GPS stations (Nahmani, 2012). Concerning the propagation delays, the ionosphere-free phase combinations are corrected for second- and third-order ionospheric refraction terms (Hernandez-Pajares et al., 2007; Petrie et al., 2010). For tropospheric modeling, we use the VMF1 mapping functions and a priori hydrostatic delays derived from the ECMWF model (Boehm et al., 2006b). The zenith wet delays (ZWD) and total tropospheric horizontal gradients are estimated with a sampling of 1 h and 24 h, respectively. During the least squares estimation, the ZWD is represented by a piecewise linear function with hourly nodes modeled as a random walk, i.e. $ZWD(t + \Delta t) = ZWD(t) + \varepsilon(t)$. The variance of the white noise $\varepsilon(t)$ is parameterized as $\sigma_\varepsilon^2 = q_{rw}^2 . \Delta t$, where $\Delta t = 1h$ and the parameter $q_{rw}$ is fixed to the recommended value of 20 mm.h$^{-1/2}$ (Herring et al., 2015). To avoid edge effects on the ZWD estimates, we use a sliding window technique. We process GPS data in two 24-h sessions, the first starting at 00:00 UTC and the




second at 12:00 UTC, from which the central 12 h output estimates were extracted as the final solutions. At the end of each session processing, the estimated tropospheric delays, tropospheric gradients and the GPS one-way phase residuals (Herring et al., 2015) are saved for each AMMA station (Table 1).

The processing of GPS data with the GAMIT software has some limitations: the ZWD temporal sampling of

1 h and the gradient temporal sampling of 24 h are too coarse to document the rapid meteorological extreme events like the passage of MCSs. Indeed, in such situations, rapid variations in the ZWD and strong anisotropy in the refractivity field are observed (Bock et al., 2008). As a consequence of the severe mis-modeling of the propagation delays, the estimated ZWD and gradient parameters are likely biased. Inspection of the post-fit phase residuals will help identify the severity of the mis-modeling. This mis-modeling can be partly overcome

by the use of a PPP Kalman-filter based software to increase the sampling of ZWD and gradients estimates. The PPP approach has also the advantage of eliminating the propagation of errors induced by the mis-modeling between stations.

### 2.2    GPS Data Analysis with GIPSY-OASIS in PPP mode

The GIPSY-OASIS II v 6.3 software is based on a Kalman-filter estimator (Zumberge et al., 1997). We used it in

PPP mode to reprocess the AMMA GPS data with a parameterization as similar as possible to the settings used with the GAMIT software described in Section 2.1. Table 1 summarizes the different strategies used to process the GPS data. Compared to GAMIT, a major difference is the sampling interval of the estimated parameters (ZWDs, tropospheric gradients and receiver clocks) of 5-min which is achievable thanks to the Kalman-filter technique in which the parameters are updated at each time step. A minor drawback is that the GPS phase

observations are thus decimated to a 5-min interval, i.e. this estimation procedure uses 10 times fewer observations than the GAMIT processing. A few other minor differences are in the observation weighting function which is modeled as $\sigma^2 = a^2 / \sin^2(e)$ with $a$ fixed to 10 mm and the phase ambiguities resolution methods (GIPSY-OASIS II v6.3 uses the "ambigon" algorithm of Bertiger et al. (2010)). To avoid edge effects, we followed the standard procedure used at JPL. The GPS data are analyzed in a 30 h window centered on each

25   day from which the 00:00-24:00 UTC parameters are extracted. This is also consistent with the JPL satellite orbit and clock products which are used instead of the IGS products. Regarding tropospheric modeling, ZWD and tropospheric gradient parameters are modeled as random walk processes with a 5 min time resolution. The question arises as to the choice of the parameterization of the random walk. The recommendations in the GIPSY-OASIS II documentation are mainly based on the results of Bar-Sever et al. (1998) and updated by Selle and

Desai (2016). They suggest that the parameter of the ZWD random walk should be set to at least 8.4 mm.h$^{-1/2}$ when using $\sigma^2 = a^2 / \sin^2(e)$ as weighting function for the GPS observations (against 20 mm.h$^{-1/2}$ in the GAMIT software). However, Jarlemark et al. (1998) noticed that using a random walk parameter tighter than real atmospheric variability introduces errors in ZWD estimates. Since this study is focused on MCSs which correspond to severe weather events, we can expect a very high variability of tropospheric parameters and it

seems more reasonable to set the random walk parameter to the looser recommended value in both software rather than to the tighter. As for the temporal evolution model of tropospheric gradients, it is common to use a random walk whose parameter ten times smaller than that of ZWD (Bar-Sever et al., 1998). Thus, we keep the



GAMIT-recommended random walk parameter of ZWD to 20 mm.h$^{-1/2}$ and set the random walk parameter of tropospheric gradients to 2 mm.h$^{-1/2}$.

### 2.3 Comparison and quality of the GPS Solutions from GAMIT and GIPSY-OASIS software

#### 2.3.1 ZWD and gradients

The time series of ZWD retrieved by the GAMIT software at Timbuktu, Niamey, and Djougou, between 2006 and 2008, display a well defined annual cycle (Fig. 2). The dry seasons are characterized by ZWD less than 100 mm (ca. between December and March) while the wet seasons, also referred to as the monsoon seasons, between June and September (hereafter, JJAS), are characterized by frequent rain events and ZWD higher than 200 mm (Bock et al., 2008). The duration of the monsoon season shrinks as one goes from the wetter climate in the South

to the more arid climate in the North.

       Table 2 presents the statistics of the differences of ZWD estimates (ΔZWD) obtained with both software, during the monsoon season. The average differences of ZWD range from -0.45 mm at Gao to 0.96 mm at Djougou. They are small but statistically significant at all stations (95% confidence level) and might be explained by minor differences persisting between the two GPS data processing (differences in orbits, tropospheric modeling, weighting of observations and potential rejection of some of them). Standard deviations

of ΔZWD range from 3.69 mm at Tamale to 4.30 mm at Gao. To understand the origin of these differences, we made a further statistical analysis of the tropospheric estimates presented in Fig. 3 for ZWDs and Fig. 4 for gradients at Niamey. Figure 3a shows that the time series of ΔZWD exhibits a greater dispersion during the wet season than during the dry season as expected from larger (smaller) biases and mis-modeling of tropospheric refractivity with GAMIT during the wet (dry) season. To better characterize the time scales of atmospheric

variability we computed sub-daily and sub-hourly variability estimates. To quantify the sub-daily (i.e. diurnal) variability, we subtracted a 1-day moving average from the hourly time series and then computed daily standard deviations. The sub-hourly variability was quantified in a similar way by subtracting the hourly time series from the 5-min time series. Figure 3b and 3c show that the sub-daily and sub-hourly variability in ZWD estimated

with GIPSY-OASIS dramatically increases during the wet season. The GAMIT solution presents very similar behavior for the sub-daily variability (not shown), but careful inspection of the 30-day smoothed time series in Fig. 3b shows that the GAMIT solution exhibits slightly larger sub-daily variability than GIPSY-OASIS during the wet season. This can be explained by random ZWD biases induced by mis-modeling. Indeed, the tropospheric modeling used in the GAMIT software does not take into account the sub-daily variations of the

gradients nor the sub-hourly variations of ZWD. The effect is therefore more pronounced during the wet season because of higher tropospheric variability in these parameters as shown in Fig. 3c and Fig. 4 (and further analyzed on a case study in Section 3).

       Tropospheric gradients show a strong seasonality in phase with the WAM (Figs. 4a, b). The daily averages of gradients are in good agreement between the two software (not shown) with a mean difference smaller than a

one-tenth of a millimeter and a standard deviation smaller than 0.22 mm due to biases induced by the un-modeled sub-daily variability. The sub-daily variations of the gradients estimated with GIPSY-OASIS can be quite large (up to 2 mm), with an average standard deviation of around 0.6 mm in both components, during the


wet season (Figs. 4c, d). The sub-hourly variations of gradients are on the other hand quite small with an average standard deviation of 0.12-0.15 mm (Figs. 4e, f).

At 7° of elevation, an uncertainty of the order of 1.5 mm on the ZWD during the JJAS periods (see Fig. 3c for the 1D-STD of "5-min ZWD – $MA_{1H}$") leads to an uncertainty of 12 mm on the slant tropospheric delay (S-Trop-D) using the GMF (Boehm et al., 2006a). Failure to take into account the hourly variations of the gradient of the order of 0.6 mm leads to an uncertainty on the S-Trop-D at 7° of elevation of around 33 mm using the specific mapping function of Chen and Herring (1997). Neglecting the sub-hourly variations of the gradient (of the order of 0.2 mm) leads to an uncertainty of about 11 mm on the STD. Thus, the global uncertainty on the S-Trop-D induced by these defects of tropospheric modeling in the setting of the GPS data processing with the GAMIT software is evaluated at 37 mm on the S-Trop-D (i.e. 4.7 mm on ZTD if projected back to zenith using the GMF mapping function) during the wet season. During DJFM periods, the global uncertainty on the S-Trop-D is evaluated at 16 mm (i.e. 2 mm on ZTD) with 0.9 mm, 0.25 mm and 0.09 mm as uncertainties on ZWD, hourly and sub-hourly variations of the gradient respectively. Tropospheric mis-modeling map onto the GPS phase residuals and its effects are moduled by the season what leads to a seasonal variation of the Weighted Root Mean Square (WRMS) of the post-fit phase residuals.

### 2.3.2 WRMS of GPS phase residuals

Post-fit phase residuals of the ionosphere-free observations are traditionally inspected to check the quality of GPS observations and models. Usually, the quality of the data processing of each session (24h or 30h) is summarized in the WRMS of all residuals of the session which is computed by the software. With GAMIT, the daily WRMS of phase residuals fluctuates around a median of 12.1 mm with a median absolute deviation (MAD) of 2.4 mm, over the entire period and all stations. Over the same period, with the GIPSY-OASIS software, the median is 10.4 mm with a MAD of 1.1 mm. The statistics are very similar, indicating that on average the solutions from both software are equivalent, though the results suggests that the data processing implemented in GIPSY-OASIS is slightly more accurate.

We know from the previous sections that the mis-modeling of tropospheric variability with GAMIT is exacerbated during the wet season. Figure 5 shows the time series of daily WRMS of post-fit phase residuals at three stations for both software. The contrast in WRMS between the wet and dry season is very marked in GAMIT at all three sites (e.g., 12 mm vs. 8 mm on average at Timbuktu). The use of only one tropospheric gradient per session of 24 hours during the wet season is less suitable for tropical stations: the unmodeled tropospheric anisotropy is thus mainly reflected by the behavior of the GPS phase residuals. With GIPSY-OASIS, there is a small seasonal variation as well, indicating that during the wet seasons some modeling defects may persist (e.g. higher order tropospheric anisotropy not modeled by tropospheric gradient parameters and signal scattering from the ground, so called multipath, which depends on soil moisture (e.g. Larson et al., 2008)). During the dry season, the WRMS from GAMIT and GIPSY-OASIS are equivalent at the Sahelian stations (Timbuktu and Niamey in Fig. 5) which indicates that the tropospheric model parameterization used in GAMIT (1 ZWD every hour and 1 gradient every 24h) is adequate.

Overall, the analysis of the annual cycle presented in Section 2 emphasizes a strong increase of the sub-daily variability of GPS tropospheric estimates (ZWD and gradients) and post-fit phase residuals during the wet





season, when the atmosphere is moister and MCSs are observed. In the following section the analysis of a case study allows to shows how such MCSs generate variability at this fine time scale.

## 3 Case study of MCS over Niamey, Niger on 11 August 2006

The MCS that passed over Niamey (Niger) on 11 August 2006 is a typical case of an intense weather event in the
Sahel associated with the WAM. It has been the subject of a comprehensive study by Chong (2010) - see also Risi et al. (2010) - and is well documented by radar and meteorological data. Here, the relevance and limitations of the GPS tropospheric estimates (ZWD and gradients) and post-fit phase residuals during the passage of this MCS are investigated.

The MCS formed over the border between Chad and Nigeria on the evening of 9 August 2006. It propagated
westward, intensified over Nigeria on 10 August and reached Niamey at 03:15 UTC on 11 August. It was observed from satellite imagery in Fig. 1 approaching Niamey from the East. Figure 6 shows reflectivity maps from MIT's C-band Doppler radar at 2:10 UTC (a), 2:40 UTC (b), 3:30 UTC (c), 4:40 UTC (d) and 5:40 UTC (e) that illustrate the dynamic of the MCS organized into a commonly observed north-south oriented squall-line. The two parts of the MCS are easily identifiable: the convective part with a reflectivity above than 37 dBZ (in
yellow, orange or red) and the stratiform part, more extended, with a reflectivity lower than 37 dBZ, mainly in green. The GPS phase residuals superposed in the figure are discussed later in subsection 3.3.

### 3.1 Meteorological surface data

Figure 7 shows surface meteorological data retrieved from Atmospheric Radiation Measurement (ARM)-Mobile Facility (ARM-MF) in Niamey (Miller and Slingo, 2007) at 1-min sampling (black lines) and from PTU200
sensor at 15-min sampling (red dashed lines) between 00:00 UTC and 08:00 UTC. These sensors are collocated with the AMMA GPS station on the Niamey airport site. The ARM-MF data are preferentially used for this case study because of their accuracy and their higher temporal frequency.

Time series of wind speed and direction (Fig. 7a) allow to precisely detect the arrival of the gust-front preceding deep precipitating convective cells (e.g. Largeron et al., 2015; Lothon et al., 2011). Provod et al.
(2016) associated the cold pool crossing time (CPCT) with the time of a sudden wind direction change happening within 5 min or less and reaching at least 30° in magnitude. Between 00:00 UTC and 03:03 UTC, the wind speed varies between 1.5 m/s and 3m/s with a few peaks (not exceeding 4 m/s) and a direction ranging from 200° to 250°. Between 03:03 UTC and 03:18 UTC, it is around 2 m/s, then drops to less than 1 m/s at 03:06 UTC, and suddenly jumps to 5.81 m/s at 03:18 UTC while the wind direction changes from 200° to 110°.
The gust-front is detected at 03:16 UTC (Chong, 2010) and the first rainfall is recorded 18 minutes later at 03:33 UTC at the arrival of a mature deep convective cell. During this period, the wind speed is significantly higher and ranges from 4 m/s to 6 m/s, then slightly weakens, a few minutes before the first rainfall. The surface temperature rapidly drops, at first from 26.5°C to 24.5°C (Fig. 7b). Similarly, the relative humidity suddenly and very briefly drops from 74% to 70% within 2 minutes at the CPCT (Fig. 7c). This time sequence is very typical
of the high-frequency fluctuations of surface meteorology observed with the passage of an MCS (e.g. Zipser, 1977). Then, relative humidity increases again to reach a maximum of 76% before the first rainfall.



From 03:33 UTC until 06:41 UTC, the MCS event is characterized by a two-phase rainfall pattern covering a period of 182 minutes. Between 03:33 UTC and 04:20 UTC approximately, the convective part of the MCS produces heavy convective showers greater than 10 mm/h over a cumulative period of 29 minutes and exceeding 20 mm/h over a cumulative period of 12 minutes. At the beginning of the first shower, a second significant drop

in surface temperature, from 24.5°C to 22°C between 03:32 UTC and 03:46 UTC is observed (Fig. 7b). Likewise, the relative humidity sharply increases to about 92%. From around 04:20 UTC, the trailing part of the MCS produces stratiform rainfall with much lower intensity and much longer duration than convective rainfall: lower than 1 mm/h over a cumulative period of 94 minutes. The wind speed is then weaker (between 2 m/s and 4 m/s), before it drops to almost 0 m/s, which, with the cessation of rainfall, marks the end of the MCS event.

Surface pressure also fluctuates during the three major steps of the MCS's life: it increases during the minutes preceding the formation of the convective cell above Niamey; it presents a local maximum during the convective phase and then it slightly decreases during the stratiform phase. The surface pressure sequence is again consistent with previous studies (e.g. Redelsperger et al., 2002).

Figures 7b, c, and d show that the data retrieved from the PTU200 sensor are in good agreement with data

from ARM-MF but the 15-min sampling is not sufficient to detect that the surface temperature drops in two consecutive stages or the sudden and brief drop in relative humidity. There is thus really added value of having the high sampling data from ARM-MF to capture details of the internal dynamics of the MCS. However, such data are only available at Niamey (Niger) for 2006 only. The best way to identify the CPCT with PTU200 data at the other AMMA GPS stations is thus to detect significant drops in surface temperature over a period between

30 minutes to 1 hour followed by strong rainfall (see section 4).

### 3.2    GPS estimates

The GPS tropospheric estimates retrieved from the GIPSY-OASIS (black lines) software allows to document the MCS with a 5-minutes sampling rate (Fig. 8). It includes the estimates from the GAMIT software (red dashed lines). In Fig. 8a, ZWD is retrieved from ZTD which hydrostatic part is computed using 1-min ARM-MF surface

pressure data and applying the formula from Saastamoinen (1972). Figure 8a also shows IWV converted from ZWD using 2-m temperature data and the formula of Bevis et al. (1992). East and north components of the tropospheric gradients are given in the Figs. 8c and 8d respectively.

Comparing the ZWD estimates retrieved from GAMIT and GIPSY-OASIS (Fig. 8a), a good agreement can be seen in both solutions, with namely a ZWD peak coinciding with the rainfall event. However, the 5-min

sampled ZWD time series from GIPSY-OASIS reveal finer time-scale information. ZWD increases from 315 mm at 01:40 UTC to 372 mm at 03:25 UTC, just a few minutes before the first rainfall (Fig. 8a). It remains above 360 mm from 03:00 UTC to 04:10 UTC, while at the same time, heavy rainfall dries out the atmosphere from 03:33 UTC onwards. Over this period, the equivalent IWV is quite stable, between 55 kg.m$^{-2}$ and 57 kg.m$^{-2}$ while around 10 kg.m$^{-2}$ of water vapor is extracted from the total column by precipitation (Fig. 8b). The

dynamics of the active convective cell involves a convergence of moisture from its immediate environment towards it. Air saturated with water leads to the significant rainfall by 37 min. The stability of IWV is well explained by a balance between moisture convergence and precipitation at first order (this implies very minor contributions of surface evaporation and water phase changes to the water budget at this scale). The equilibrium can be broken, e.g. by mixing with drier air in the immediate environment of the cell and/or by processes related





to the formation of the incipient convective cells. Such a break occurs around 04:05 UTC as shown by a decrease in ZWD followed by a decrease in precipitation intensity from 04:10 UTC onwards and marks the transition from the convective part of the MCS to its stratiform part. Between 04:10 UTC and 06:00 UTC, ZWD decreases from 360 mm to 297 mm which corresponds to a decrease of around 9.7 kg.m$^{-2}$ in IWV. Over the same period,

5    2.6 kg.m$^{-2}$ of atmospheric water is precipitated, which implies that 7.1 kg.m$^{-2}$ of water has been transported elsewhere, possibly to the newly formed convective cells. The troposphere is thus cooler and drier after the passage of the MCS.

Tropospheric gradients retrieved from the GIPSY-OASIS software show the east-west motion dynamic of the MCS between 01:00 UTC and 07:00 UTC. From 01:30 UTC onwards, the east component increases from 0.26

10   mm to reaches a maximum of +2.25 mm at 03:20 UTC while the north component remains almost null over this same period. The tropospheric gradient vector actually points to the direction of the incoming convective cells and its modulus shows a first local maximum at the CPCT (Figs. 8c, d). This is the time when the anisotropy of the troposphere around the GPS station is the most pronounced (Fig. 6c). Then, the east component decreases from its maximum to about 0 at around 03:48 UTC when the GPS station is surrounded by convective cells. It

decreases to a minimum of -2.79 mm at 04:35 UTC. The modulus of the gradient vector actually shows a second maximum at the end of the motion of the convective zone through the GPS station, when, once again, the anisotropy of the troposphere around the GPS station is the most pronounced (Fig. 6d).At around 05:40 UTC, anisotropy of the stratiform part of the MCS is detected in the northeast direction by gradients, which is confirmed by reflectivity maps (Fig. 6e). Thus, the tropospheric gradients retrieved from the GIPSY-OASIS

software appear to be relevant to contribute to the description of the dynamics of intense weather events such as MCS. Tropospheric gradients retrieved once per 24-hour session from the GAMIT software provide no such information but the examination of the post-fit phase residuals is more informative in this case.

### 3.3    GPS phase residuals

The daily WRMS computed by the software for the sessions covering the 11 August 2006, are 15 mm with

GAMIT and 10 mm with GIPSY-OASIS (see sub-section 2.3.2). In this sub-section, we analyze in detail the behavior of the phase residuals during the case study. To do so, Fig. 9 shows the RMS of the GPS phase residuals computed at 15-min sampling (a), the number of GPS observations per epoch (b) and the GPS phase residuals (c) retrieved from the GIPSY-OASIS (black) and the GAMIT (red) software during the MCS event.

With GIPSY-OASIS, the 15-min RMS of the GPS phase residuals exceeds 15 mm between 03:10 UTC and

04:35 UTC during the crossover of the convective cells. It indicates that the tropospheric anisotropy is more complex than can be inferred from gradients, which is confirmed by the GPS phase residuals on the reflectivity maps (Figs. 6c). Indeed at 03:30 UTC, the GPS phase residuals exceed 25 mm in the northeast quadrant, are below -10 mm in the northwest quadrant and between -7 mm and 7 mm elsewhere. Before and after this period between 03:10 UTC and 04:35 UTC, the RMS generally oscillates between 5 and 10 mm. Otherwise, the RMS

can be slightly higher due to very few mis-modeled and not deleted GPS observations which produce residuals over 30 mm on absolute value (Fig. 9c –black dots). In general, the behavior of the GPS phase residuals reflects the part of the tropospheric anisotropy that cannot be captured by the gradients. It comes out as spatial correlations on their projected skyplots (Figs. 6 & 10) and is reflected as temporal correlations in their time series (Fig. 9c). Increased scatter in the GPS phase residuals retrieved from GIPSY-OASIS is clearly visible





during the MCS event (Fig. 9c) but the 5-min sampling of the GPS data does not make easy to detect their temporal correlations. The number of GPS observations per epoch is usually over 9 and decreases slightly during the entire event but remains above 7 (Fig. 9b). The small number of observations that are rejected indicates that the chosen settings of the GPS data processing with the GIPSY-OASIS software appear to be adequate to study

this kind of extreme weather event.

With the GAMIT software, the RMS of the GPS phase residuals computed at 15-min sampling oscillates between 4 mm and 8 mm before and after the MCS event (Fig. 9a). It skyrockets as soon as a sufficient number of GPS observations cross the MCS from 8 mm to 20 mm at 02:00 UTC to then it oscillates between 11 mm and 22 mm as long as the MCS affects the GPS data. The RMS decreases from over 20 mm at 05:25 UTC to 5 mm at

05:50 UTC while the stratiform part of the MCS is still above the GPS station. Thus, we can assume that the GPS observations that cross the convective part of the MCS are the most disrupted and therefore be poorly modeled during the GPS data processing which leads to higher phase residuals and therefore to higher RMS. Further understanding of the RMS variation while the MCS affects the GPS data is delicate as up to 50% of the observations per epoch can be rejected as outliers (Fig. 9b – red line). This loss of data has to be put into

perspective: the 30-second sampling of the GPS data processed by GAMIT leads to process 10 times more data than with GIPSY-OASIS and thus much more residuals are available for study (Fig. 9c – red dots). Strong deviations in the phase residuals are clearly visible during the MCS event on some satellites as revealed by temporally correlated patterns.

We thus examined the projected GPS phase residuals from GAMIT and superimposed on the reflectivity

maps retrieved from MIT's C-band Doppler radar (Fig. 10).

Assuming a tropospheric thickness of 10 km and a spherical Earth of 6400 km of radius, the GPS phase residuals are projected on horizontal plane of the local coordinate system centered on the GPS station to be approximately superimposed on the reflectivity maps retrieved from MIT's C-band Doppler radar. The results obtained at 02:10 UTC (a), 02:40 UTC (b), 03:30 UTC (c), 04:40 UTC (d) and 05:30 UTC (e) are given by Fig.

6 for GIPSY-OASIS and by Fig. 10 for GAMIT.

At 02:10 UTC, the GPS phase residuals exceed 20 mm in the southeast direction because the associated GPS observations are disturbed by the top of the MCS's anvil that can not be detected by the radar waves at 0.7° elevation. Between 02:10 UTC and 03:00 UTC, the GPS phase residuals within a radius of 25 km are in an acceptable range but beyond 25 km, those located to southeast indicate the presence of convective cells and

exceed 20 mm while those located to other directions are below -20 mm (Fig. 10b). At 03:30 UTC, convective cells are above the GPS station and GPS phase residuals are very disturbed within a radius of 25 km while those beyond 25 km located to east or west are below -20 mm (Fig. 10c). At 04:40 UTC, convective cells are located west of the station. The GPS phase residuals within a radius of 25 km are negative but stay in an acceptable range while those located to the convective cells always exceed 20 mm. At 05:40 UTC, the GPS phase residuals

located to the northwest exceed 20 mm which explains the signal detected on the north component of the tropospheric gradients in the previous subsection. Thus, with the GAMIT solution, the tropospheric anisotropy induced by the MCS is mainly reflected in GPS phase residuals and the east-west crossing of the MCS can be detected in the temporal evolution of the extreme phase residuals in the maps of the Fig. 10. However, for a more quantitative study, the tropospheric gradients retrieved by GIPSY-OASIS would be more accurate.



## 4 Composites of surface variables and GPS estimates during MCS events over West Africa

Section 3 provided a detailed analysis of a single case study. Here, we aim to characterize the West African MCS properties in a systematic way using the data from the six AMMA GPS stations over three monsoon seasons, between 2006 and 2008.

### 4.1 Detection of MCS at AMMA GPS stations

A crucial step is to define a method to detect properly MCSs based on surface meteorological data. Provod et al. (2016) used 1-min averaged pressure, temperature, and wind observations from the ARM-MF in Niamey to detect the arrival of cold pools associated with MCSs, which they subjectively verified from the MIT radar and/or Meteosat satellite images. They detected 42 cold pools at during the Special Observing Period (SOP) of the AMMA project (1 June–30 September 2006). Among them, 33 were squall-line MCSs, 4 were non-squall-line MCSs, one was from a freshly dissipated MCS and 4 were from local non-MCS convection.

As in Provod's study (2016), here we use a method of detection of cold pools based on surface meteorological data. These data, available from the PTU200 sensors attached to the AMMA GPS stations include pressure, temperature, and relative humidity (recorded with a sampling rate of 15 minutes), but no wind data. First, we will primarily detect rainfall events and then select those which show a surface air temperature drop characteristics of cold pools arrival ahead of the MCS's peak rain rate.

The quality of the PTU200 data was first assessed by comparison with the ARM-MF data at Niamey. The mean ± one standard deviations of differences in pressure, temperature, and relative humidity were $0.0 \pm 0.1$ hPa, $0.4 \pm 0.8$ °C and $0.6 \pm 1.9$ %, respectively. The rain rate data were retrieved from the 3-hourly, 0.25 by 0.25° gridded precipitation data (3B42 v6) of the Tropical Rainfall Measuring Mission (TRMM) Multisatellite Precipitation Analysis (Huffman et al., 2007). The TRMM rain rates were compared point to point to the ARM-MF data at 3-hour sampling (2972 points). Rainfall is detected on 243 points by either TRMM or ARM-MF data but only on 74 points in both data sources. On 169 remaining points, rainfall is observed either only by TRMM but not by ARM-MF or the opposite. In 145 points out of 169 (86%), the rain rate is less than 0.5 mm.h$^{-1}$. The 16 remaining points (14%) correspond to more intense rainfall but observed at different times. This result led us to define a rainfall event as an uninterrupted period of precipitation during which the maximum rain rate is larger than 0.5 mm.h$^{-1}$.

We detect 27 events using the 3-hour ARM-MF rainfall data during JJAS 2006 and 47 events using the TRMM rainfall data. Among the 27 ARM events, 25 coincided with at least one TRMM rainfall event. The two ARM events not detected by TRMM are of very low intensity and correspond most probably to small local showers. Each event is associated with a start time, a time of maximum precipitation, and an end time. If several rainfall events occur within a 13-hour window they are merged together as we will be mainly interested in MCSs events in which several convective cells can be detected. We keep the rainfall events with average precipitation above 0.5 mm.h$^{-1}$ and with a peak rain rate above 1 mm.h$^{-1}$. This method detects 27 events from the TRMM data and 20 events from the ARM-MF rainfall data. Compared to the first intercomparison, the agreement between the two datasets is thus improved when considering longer and more intense events. The seven TRMM events that do not match with any ARM-MF rainfall event may be due to the difference in spatial representativeness of rainfall between the two datasets: very localized for ARM data and more extended for TRMM data (0.25 ° ~ 30



km). Since the GPS tropospheric estimates are sensitive to weather events occurring within a radius of approximately 75 km around the station (as illustrated Figs. 6 and 10), we expect actually better agreement with the TRMM observations than with local precipitation observations.

For the detection of MCSs, we identify the CPCT by seeking and dating the greatest temperature drop over 1

5    hour within a time window defined from the event start time minus 4 hours to event end time plus 3 hours. To detect a cold pool associated with an MCS, the temperature drop must be at least of 1°C as in Provod et al. (2016).

Figures 11 and 12 show the events detected using our method at Niamey data during August 2006 along with surface meteorological observations from the PTU200 sensor and ARM-MF data (Fig. 11) and GPS tropospheric

estimates (Fig. 12) retrieved with both software approaches (GIPSY-OASIS as black lines and GAMIT as red dashed lines). The method detects 10 MCS events based on the rain rates from TRMM data (Figs 11c and d) and temperature from the PTU200 (Fig. 11b). In general, all the detected MCS cases were confirmed from the reflectivity maps of the MIT C-band Doppler radar. One event (14 August 2006) is not detected in the ARM-MF rain rate data. It is actually a propagating cold pool from a freshly dissipated MCS (Provod et al., 2016). All

spotted periods include a sudden increase in the wind speed associated with a sudden change of direction (Fig. 11a), which are the characteristics of the arrival of a cold pool. They also include a pronounced and sudden drop in temperature (Fig. 11b), an increase in relative humidity (Fig. 11c) and in ground pressure (Fig. 11d). This is all consistent with previous studies (e.g. Redelsperger et al., 2002). There is also a clear peak in ZWD (Fig. 12a), a signal in tropospheric gradients retrieved from GIPSY-OASIS (Figs. 12b and 12c) and an increase in 15-min

RMS of GPS phase residuals retrieved from GAMIT (Fig. 12d). Careful inspection of the wind, temperature, tropospheric gradients and phase residuals reveal two potential events with cold pool-like signatures on 30 and 31 August that were not detected because there was no signal in rainfall. One can notice that there is no peak in the ZWD data associated with these potential events (Fig. 12a). Inspection of the MIT radar data actually shows decaying MCSs for these two cases.

Thus, of the 42 cold pools detected and categorized by Provod et al. (2016), the detection procedure retains the 24 most intense ones including large MCSs, mainly squall-lines, over the JJAS period in 2006. The MCS detection procedure was then applied to the six AMMA GPS stations over the JJAS periods from 2006 to 2008.

## 4.2    Composites of meteorological variables and GPS estimates during MCS events in JJAS 2006 to 2008

With the detection method described in section 4.1, we flagged a total of 414 MCSs passing over the six AMMA GPS stations in JJAS 2006 to 2008. The detailed characteristics for each event (start, peak, and rainfall end times, CPCT, temperature drop and precipitation peak and cumulative rainfall depth) are given in the supplemental material. The list includes only events for which both PTU200 and GPS data were available within the 10-hour windows centered on the CPCTs (i.e. a few events may not be documented because of gaps in our

data).

Table 3 reports the number of events and statistics of surface air variables, precipitation, and ZWD recorded by the six stations. The meridian climatic gradient between the Sahara and the Guinean coast is reflected both in the number of MCS events and the atmospheric variables. The number of events ranges from 31 in Timbuktu to 116 in Djougou, i.e. in a ratio of 1:4. Note that the number of events detected at Tamale is smaller due to failures





with the GPS receiver in 2007. The atmospheric environment of these MCSs is characterized with surface air temperature, pressure, and relative humidity, and ZWD. Again, the meridional climatic gradient is clearly reflected in these variables. Surface air temperature and relative humidity range from the very hot (over 34°C) and dry (below 45%) saharo-sahelian climate to milder (below 30°C) and moister (over 69%) sudano-guinean

climate. The mean surface pressure is not very informative here as it mainly reflects the altitude of the station, but variability is seen to be stronger at the sahelian sites (Niamey, Gao and Timbuktu). A contrast is also seen in the ZWD values which are below 270 mm at Timbuktu and Gao and above 290 mm at the other sites. ZWD characterizes the total column water vapor which is, as expected, lower on average and with a stronger variability in Mali than at southernmost sites. However, the relative contrast between the ZWD values is not as

strong as between the surface relative humidity values. Surface humidity is strongly controlled by surface evapotranspiration (which is a more limiting factor due to relatively low soil moisture content and vegetation cover in the Sahel (Lohou et al., 2014)) while ZWD is largely influenced by upper level moisture transport (which is more of large-scale nature – e.g. synoptic-scale fluctuations induced by African easterly waves (Barthe et al., 2010) can be quite large at the northernmost sites). In the south, the lower saturation vapor pressure due to

relatively cooler air is also a liming factor both to surface humidity and ZWD. In line with Lebel et al. (2003) and Frappart et al. (2009), rainfall characteristics show also marked latitudinal variations, with a minimum cumulative rain around 10 mm at the saharo-sahelian stations (Timbuktu and Gao) and a maximum at the soudano-sahelian stations (16 mm and 21 mm at Niamey and Ouagadougou, respectively). Interestingly, the saharo-sahelian stations show the most important surface air temperature drop (around -8°C in 1 hour on

average) which reflects the strong cooling (and moistening) of the cold pool air (Lothon et al., 2011; Provod et al., 2016) when it intrudes the typically warmer and drier boundary layer. Figs. 13 and 14 show composites of the surface meteorological variables, ZWD, and tropospheric delay gradient components, centered on the CPCT of the MCSs for all six AMMA stations. Overall, the variables show similar temporal variations at all the sites. The variations are also highly consistent with those observed for the 11 August case study presented in Section

3: a drop in surface air temperature coincident with a jump in relative humidity and in surface pressure, as well as a gradual increase in ZWD peaking slightly after (~30 min) the CPCT. The east gradient component also shows a systematic oscillation across the CPCT, while the north gradient is more stationary throughout the time window. There is a clear increase of the magnitude of all changes around the CPCT with the latitude of the site (along the south-north climate gradient). Surface air temperature (Figs. 13a and 14a) and relative humidity (Figs.

13b and 14b) do not fluctuate much before and after the CPCT. The magnitudes of relative humidity and surface pressure jumps are coupled with the magnitudes of the temperature drops. They reach over +30% RH and +2 hPa at Gao and Timbuktu, compared to +15-20% and +1 hPa only at Djougou and Tamale. Small trends in these variables can actually be seen after the CPCT at the 4 northernmost sites which reveals a gradual drying and warming of the surface air after MCS's passage.

Figure 14d shows that the ZWD starts to increase five hours before the CPCT at a common rate of around 3 mm.h$^{-1}$. In the last 30 min before the CPCT, this rate increases, and it is only then that it depends on the latitude of the station. The smallest rates are of 8 mm.h$^{-1}$ at Djougou and Tamale, and the largest of 16 mm.h$^{-1}$ at Niamey, Gao and Timbuktu. ZWD peaks at ~30 min after the CPCT at all sites. The change in ZWD is thus strongly station (latitude) dependent. It is more marked at the northernmost stations where it reaches around +40

40    mm (relative to a reference value taken 2h before the CPCT), +30 mm at Ouagadougou and only +20 mm for the





southernmost sites. Moisture convergence associated with the propagating MCSs is thus larger in the initially drier saharo-sahelian atmospheres than in the moister Guinean region. At CPCT plus 5 hours, ZWD has generally decreased back to or slightly below its initial value. Though there is a strong moisture convergence associated with the passage of the MCS, the tendency after it is a slightly drier air column, especially in the more arid climate (this is consistent with typical signatures found in sounding data, not shown). The ZWD peak is also narrower from the southern to the northern sites, which suggests faster MCS in the north (again consistent with existing studies, e.g. (Maranan et al., 2018)).

The isotropic vision of water vapor changes in the atmospheric column described by the ZWD time series is complemented by tropospheric gradients which reflect the horizontal displacement of MCSs around the GPS stations. The east component (Fig. 13e) has a very similar behavior at all AMMA GPS stations. It is very low at CPCT minus 5 hours and slightly increases to show a maximum at the CPCT when the anisotropy of the troposphere around the GPS station is the most pronounced. Then it decreases rapidly to present a minimum in one to two hours following the CPCT which corresponds to the east-west crossing of the convective part of the MCSs. After this minimum, the east component tends to return to a value close to zero corresponding to the passage of the stratiform part of the MCSs. The behavior of the east component observed previously in subsection 3.2 seems to be common to all GPS stations in West Africa during the passage of MCSs implying that MCSs preferentially propagate in the east-west direction (which is very consistent with numerous past studies, e.g. (Mathon and Laurent, 2001)). On the other hand, the north component of tropospheric gradients (Fig. 13f) doesn't show such a systematic behavior. It would be valuable to assess whether these differences in the meridional direction agree with finding from MCS tracking (using satellite data). The stronger signal is found at Timbuktu where this component remains negative on average throughout the time window. This suggests that convective cells pass preferentially south of the station, a result which may be related to the location of the site, on the northern flank of the inter-tropical convergence zone (ITCZ). The signal is distinct and also weaker at all the other stations. It is noticeable thought that at Niamey, it is very similar to the one observed for the case study (Fig. 8), which suggests that convective cells have a slight tendency to arrive at the station from the southeast and then turn west.

## 5    Conclusions

The characteristics of GPS tropospheric estimates (ZWD and gradients) and post-fit phase residuals during the wet season (June to September) of the West African Monsoon have been evaluated using two different GPS data processing approaches. The first uses the double difference observations from a regional network of GPS stations centered on the study area (West Africa) with the GAMIT software. The second uses the undifferenced observations in PPP mode with the GIPSY-OASIS software. Hourly ZWD estimates retrieved by both strategies are highly consistent on average (RMS difference of ~4 mm). Both processing solutions exhibit a strong seasonal modulation of the ZWD and gradients variability reflecting the presence of active moisture transport during the monsoon period. In this respect, the 5-minute sampling of ZWD and gradient estimates with GIPSY-OASIS appears to be more suitable for studying rapid weather events compared to the 1-hourly ZWD and daily gradient sampling of GAMIT. Sub-daily gradient variability is shown to be quite large (1-σ of 0.6 mm on average with peaks up to 2 mm) and to represent a strong source of uncertainty in the GAMIT estimates. Mis-modelling of the



tropospheric delay variability in the GAMIT solution is reflected in the post-fit residuals during the monsoon season (the weighted RMS of residuals passes from 10 mm during the dry season to 15 mm on average during the wet season). Mesoscale Convective Systems (MCSs) are a major contributor to the rapid tropospheric delay anisotropy.

The case study of the squall line over Niamey on 11 August 2006, confirmed the good sensitivity of both GPS software to the ZWD variation associated with the passage of an MCS. In this case, a rapid increase of + 40 mm was observed in ZWD within the 2 hours preceding the rainfall peak followed by a decrease leading to a slightly lower final ZWD value (20 mm below the initial value). Thanks to the high temporal sampling of the gradient estimates, the GIPSY-OASIS solution provides additional information on the atmospheric anisotropy.

The east gradient component shows a strong oscillation reflecting the westward propagation of the MCS passing over the station. With the GAMIT solution, the horizontal anisotropy is to some extent reflected in the post-fit residuals as could be verified on reflectivity maps from MIT's C-band Doppler radar.

    These results were extended with the analysis of composites of surface meteorological variables and GPS delay estimates at all six AMMA stations over 3 monsoon seasons (2006-2008). It shows a very consistent

behavior in all variables during the time window of ±5 hours around the cold pool crossing times (CPCTs) of 414 MCS events. Qualitatively, all MCS events show a drop in surface air temperature and a jump in relative humidity and pressure at the CPCT consistent with the squall line case study and the results of Provod et al. (2016) for Niamey. Quantitatively, there is however a clear tendency in stronger magnitude of the changes at higher latitude, i.e. on the atmospheric environment specific to each site. The dry saharo-sahelian climate of Gao

and Timbuktu is prone to the steepest surface temperature drops (-9 °C) and increases in ZWD (+40 mm, i.e. about 6 $kg.m^{-2}$ IWV) due to moisture convergence close to the time when the cold pool arrives. However, cumulated rainfall (10 mm) and peak rain rate (3 $mm.h^{-1}$) are smaller there than at the soudano-sahelian sites of Ouagadougou and Niamey. This may be due to stronger evaporation of rainfall drops (Meynadier et al., 2010a) in drier climate. Furthermore at this scale, one can observe a small tendency for total column moisture to

diminish after the passage of the MCS indicating that the balance between moisture convergence and precipitation is negative while the opposite is actually seen at the moist soudano-guinean sites (Djougou and Tamale). Finally, the tropospheric gradients show the main direction of the propagation of the MCSs at all six sites is from east to west, consistent with climatology based on the satellite imagery (e. g. Laing and Fritsch, 1993).

To conclude, this study showed that the GPS tropospheric estimates are relevant for climate monitoring and documentation of intense weather events such as MCSs. They could also be used to assess the simulation of MCSs by atmospheric models.

## 6    Data availability

    GPS and PTU200 data of AMMA stations can be accessed at ftp://igs.ign.fr/ /pub/igs/data/campaign/amma (last

access 27 November 2018). ARM-MF data can be accessed at http://www.archive.arm.gov (last access 27 November 2018).



## 7 Author contributions

SN performed the data analysis (processing of the GPS data, computation of statistics) and wrote the paper. OB performed the MCS event detection based on precipitation and surface meteorological data. OB and FG contributed to the data analysis and writing of the paper.

## 8 Competing interests

The authors declare that they have no conflict of interest.

## 9 Acknowledgements

The authors would like to thank Miroslav Provod (School of Earth and Environment, University of Leeds, UK) for the details provided on his database of 42 cold pools in Niamey, Niger. ARM-MF data were obtained from the Atmospheric Radiation Measurement (ARM) Program sponsored by the U.S. Department of Energy, Office of Science, Office of Biological and Environmental Research, Climate and Environmental Sciences Division. The TRMM_3B42 data were provided by the NASA/Goddard Space Flight Center's Mesoscale Atmospheric Processes Laboratory and PPS, which develop and compute the TRMM_3B42 data as a contribution to TRMM and archived at the NASA GES DISC. This work was developed in the framework of the VEGA project and supported by the CNRS program LEFE/INSU. This work is a contribution to the European COST Action ES1206 GNSS4SWEC (GNSS for Severe Weather and Climate monitoring; http://www.cost.eu/COST_Actions/essem/ES1206) aiming at the development of the global GPS network for atmospheric research and climate change monitoring.





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



**Tables**

**Table 1: GPS data processing details with GAMIT and GIPSY-OASIS software packages**

| | GAMIT v10.6 | GIPSY-OASIS II v6.3 |
|---|---|---|
| Strategy | Network of 25 stations centered on West Africa (6 AMMA stations + 19 regional stations) <br> Free network strategy using double-difference baselines <br> 7° elevation cutoff angle <br> Data processed in two 24-h sessions (from 00 UTC to 24 UTC and from 12 UTC on day D to 12 UTC on day D+1) from which central parts are extracted (06-18 UTC and 18-06 UTC) | Stations processed individually <br> Precise Point Positioning (PPP) mode using un-differenced phase observations <br> 7° elevation cutoff angle <br> Data processed in a single 30h session (from 21 UTC on day D-1 to 03 UTC on day D+1) from which central part is extracted (00-24 UTC) |
| Orbits and clocks | IGS final orbits fixed | JPL final orbits and clocks fixed (sampled at 5 minutes dedicated to PPP mode). |
| Observation rate | 30-sec sampling | 5-min sampling |
| Observations weighting | $\sigma^2=a^2+b^2/sin^2(e)$, where $e$ is the elevation angle and $a$ and $b$ are determined from least squares adjustment residuals | $\sigma^2=a^2/sin^2(e)$ with $a$ fixed to 10 mm |
| Tropospheric Modeling | VMF1 mapping function <br> ZHD a priori from ECMWF, <br> 1 ZWD per hour constrained by a 2 cm.h$^{-1/2}$ random walk <br> 1 tropospheric gradient per 24-hour session without stochastic constraint. | VMF1 mapping function <br> ZHD a priori from ECMWF, <br> 1 ZWD per 5 min constrained by a 2 cm.h$^{-1/2}$ random walk <br> 1 tropospheric gradient per 5 min constrained by a 2 mm.h$^{-1/2}$ random walk |
| Tropospheric estimates | 1 ZWD per hour <br> 1 tropospheric gradient per 24-hour | 1 ZWD per 5 min <br> 1 tropospheric gradient per 5 min |





**Table 2: Statistics of ZWD differences from GAMIT and GIPSY-OASIS processing solutions (GAMIT minus GIPSY-OASIS) for June to September, 2006-2008. The GIPSY-OASIS ZWD estimates were beforehand averaged over 1 h intervals. The last column gives the p-value of a t-test for the difference of the means. Values smaller than $10^{-7}$ are replaced by 0.**

| Station | Mean $\Delta$ZWD (mm) | Std $\Delta$ZWD (mm) | Number of ZWD pairs | P-Value |
|---|---|---|---|---|
| TOMB | 0.69 | 4.03 | 6874 | 0 |
| GAO1 | -0.45 | 4.30 | 8609 | 0 |
| NIAM | 0.52 | 4.17 | 8588 | 0 |
| OUAG | -0.24 | 4.37 | 8170 | $9.77\ 10^{-7}$ |
| TAMA | -0.11 | 3.69 | 5731 | $2.55\ 10^{-2}$ |
| DJOU | 0.96 | 4.18 | 8720 | 0 |





**Table 3: Statistics (mean ± one standard deviation) for 414 MCS events detected at AMMA GPS stations during JJAS2006 to 2008. Characteristics of the atmospheric environment before the MCS: surface air temperature, pressure, and relative humidity from PTU200 sensors, and ZWD from GPS receivers (these variables are averaged within 5 hours before the cold pool crossing time). Cumulative rainfall depth and maximum rain rate from the TRMM Multisatellite Precipitation Analysis (Huffman et al., 2007). The last column gives the maximum temperature drop, ΔTemp, within 1 hour after the cold pool crossing time.**

| Station | Alt. [m] | Number of MCS | Temperature [°C] | Relative Humidity [%] | Pressure [hPa] | ZWD [mm] | Cumulative Rainfall [mm] | Rain rate max [mm/h] | ΔTemp [°C] |
|---|---|---|---|---|---|---|---|---|---|
| TOMB | 265 | 31 | 34.3 ± 4.1 | 43.4 ± 16.0 | 979.4 ± 1.8 | 269.8 ± 26.8 | 10.8 ± 8.6 | 3.3 ± 2.6 | -8.1 ± 3.4 |
| GAO1 | 260 | 59 | 34.7 ± 3.9 | 39.5 ± 15.9 | 979.1 ± 2.1 | 262.8 ± 29.5 | 10.4 ± 7.8 | 3.0 ± 2.3 | -8.3 ± 3.8 |
| NIAM | 224 | 74 | 30.6 ± 3.9 | 60.0 ± 16.1 | 985.0 ± 1.9 | 292.7 ± 27.6 | 16.1 ± 13.4 | 4.0 ± 3.0 | -6.5 ± 3.2 |
| OUAG | 307 | 89 | 29.8 ± 3.4 | 63.6 ± 14.8 | 976.4 ± 1.6 | 295.5 ± 26.1 | 21.7 ± 19.9 | 5.5 ± 4.8 | -6.8 ± 2.9 |
| TAMA | 172 | 45(*) | 28.6 ± 3.2 | 69.3 ± 13.0 | 991.7 ± 1.4 | 312.4 ± 20.3 | 12.6 ± 7.7 | 2.9 ± 1.8 | -5.2 ± 2.9 |
| DJOU | 439 | 116 | 26.7 ± 2.4 | 74.4 ± 11.5 | 962.6 ± 1.3 | 292.0 ± 21.3 | 15.7 ± 13.0 | 3.7 ± 3.3 | -4.6 ± 2.2 |

(*) the number of events detected at TAMA is smaller because of a long gap in the GPS data in 2007.



1 **Figures**

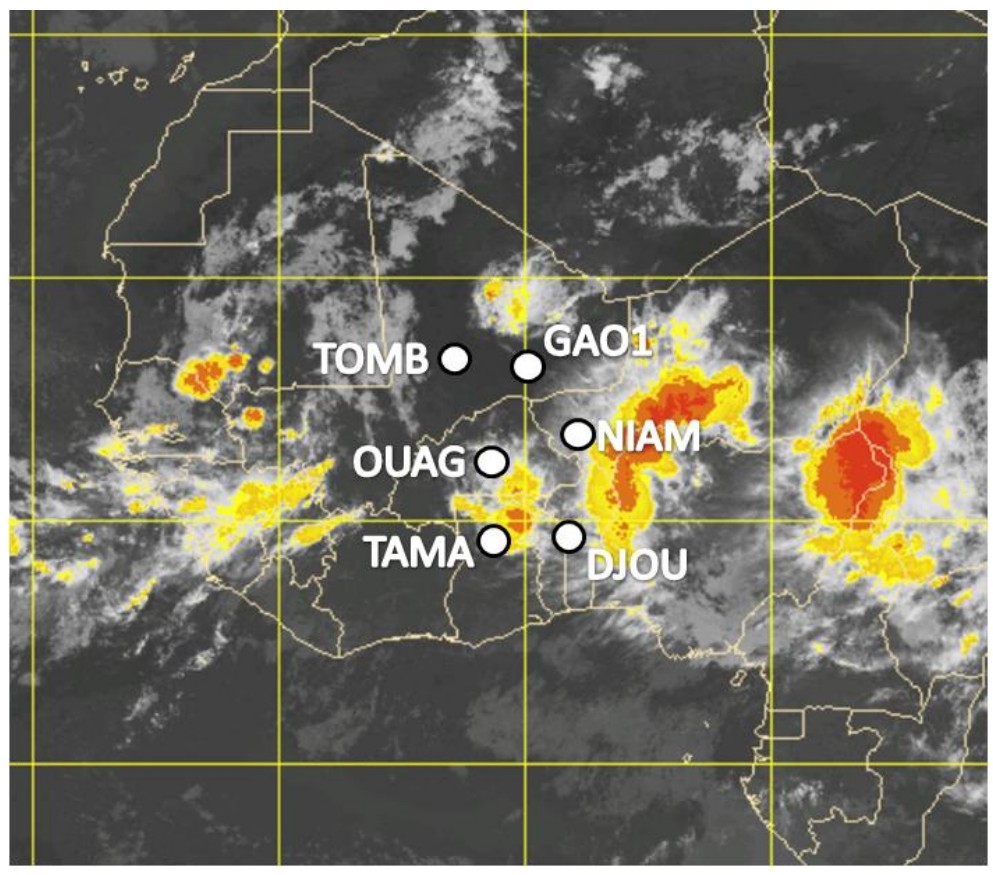

**Figure 1: Meteosat-8 infrared image of West Africa on 10 August 2006, 23:00 UTC. Brightness temperatures range between 225 K (yellow) and 195 K (red). The position of the six AMMA GPS stations are shown as white dots: TOMB (Timbuktu, Mali), GAO1 (Gao, Mali), OUAG (Ouagadougou, Burkina-Faso), NIAM (Niamey, Niger), TAMA (Tamale, Ghana), DJOU (Djougou, Benin).**





2    **Figure 2: Time series of GPS-derived ZWD retrieved with the GAMIT software and rainfall retrieved from the**
3    **TRMM Multisatellite Precipitation Analysis (Huffman et al., 2007) at (a) Timbuktu, (b) Niamey, and (c) Djougou.**
4    **The thin black solid and dashed lines superposed show 30-day moving averages and standard deviations, respectively.**
5    **The dashed black vertical lines delimit the period between June and September corresponding to the main rainy**
6    **season.**





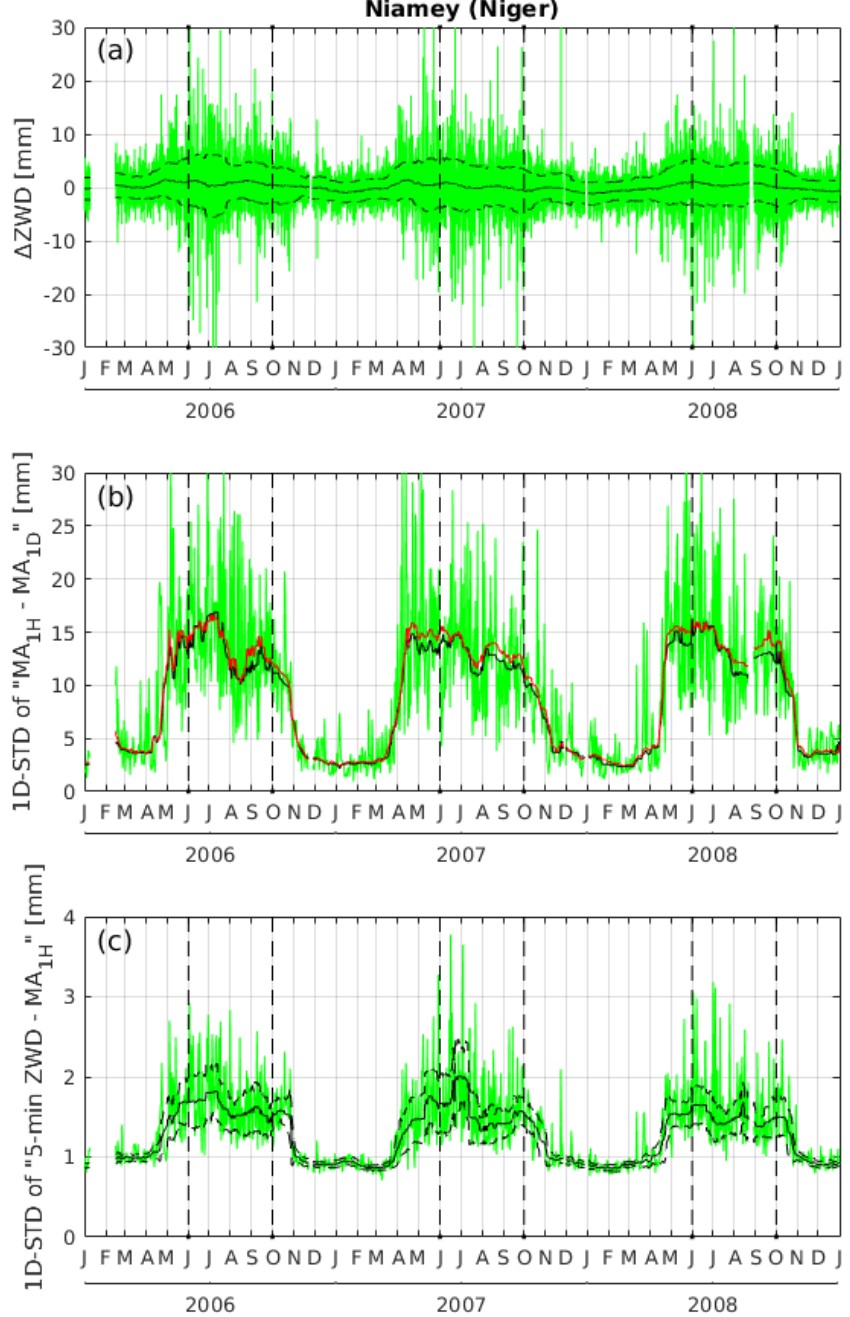

**Figure 3: Analysis of the variability of GPS-derived ZWD at Niamey. (a) Difference between GAMIT and GIPSY-OASIS hourly ZWD estimates. (b) Sub-daily ZWD variability from GIPSY-OASIS (in green: daily standard deviation of hourly minus daily mean ZWD series). (c) Sub-hourly ZWD variability from GIPSY-OASIS (in green: daily standard deviation of "5-min minus hourly ZWD series). The thin black solid lines superposed in (a), (b), and (c) show 30-day moving medians. The red line in (b) shows a 30-day moving median of the sub-daily variability computed from**





1     **the ZWD estimates retrieved by GAMIT. The thin black dashed lines in (a) and (c) show median absolute deviations**
2     **around the medians. Dashed black vertical lines indicate the JJAS period of each year.**





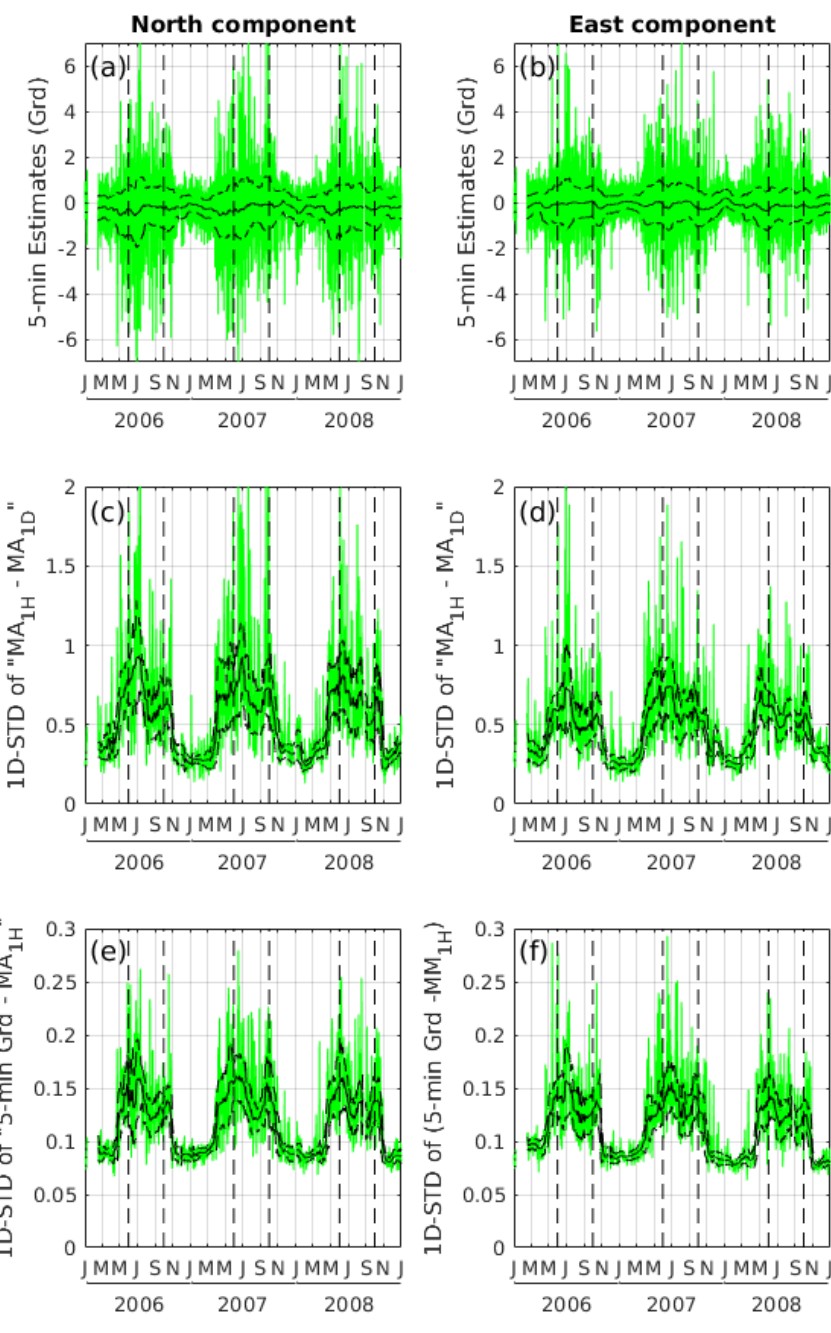

Figure 4: (a) North and (b) East components of the 5-min tropospheric gradients retrieved with the GIPSY-OASIS software at Niamey between 2006 and 2008, with 30-day moving averages shown as solid black lines and 30-day moving standard deviations around the means as dashed black lines.(c, d) sub-daily variability of North and East





1   gradients (similar as in Fig. 3b). (e, f) Sub-hourly variability of North and East gradients (similar as in Fig. 3c). In (c,
2   d, e, f) the solid black lines show 30-day moving medians and the dashed black lines show moving median absolute
3   deviations around the medians. The dashed black vertical lines indicate the JJAS period of each year.





2   **Figure 5: Daily Weighted Root Mean Square of the GPS post-fit phase residuals at (a, d) Timbuktu, (b, e) Niamey and**
3   **(c, f) Djougou. (left) results of GAMIT software and (right) results of GIPSY-OASIS software. The solid black lines**



1    show 30-day moving medians and the dashed black lines show moving median absolute deviations around the
2    medians.









1  Figure 6: Reflectivity maps from MIT's C-band Doppler radar in Niamey at (a) 2:10 UTC, (b) 2:40 UTC, (c) 3:30
2  UTC, (d) 4:40 UTC, and (e) 05:40 UTC on 11 August 2006, showing a squall line MCS approaching from the east and
3  passing straight over the site on 3:30 UTC. Superposed are post-fit phase residuals from GPS data processing using
4  the GIPSY-OASIS software at the exact times of the radar plots. The radar reflectivity color bar is in the lower right
5  angle. The GPS residual color bar is attached to sub-plot (e). The GPS phase residuals are projected on the radar
6  maps in the directions of satellite-receiver paths, assuming a 10-km thickness of the troposphere and a spherical Earth
7  of 6400 km of radius. The concentric circles indicate distance from the GPS station's location at 25-km intervals.





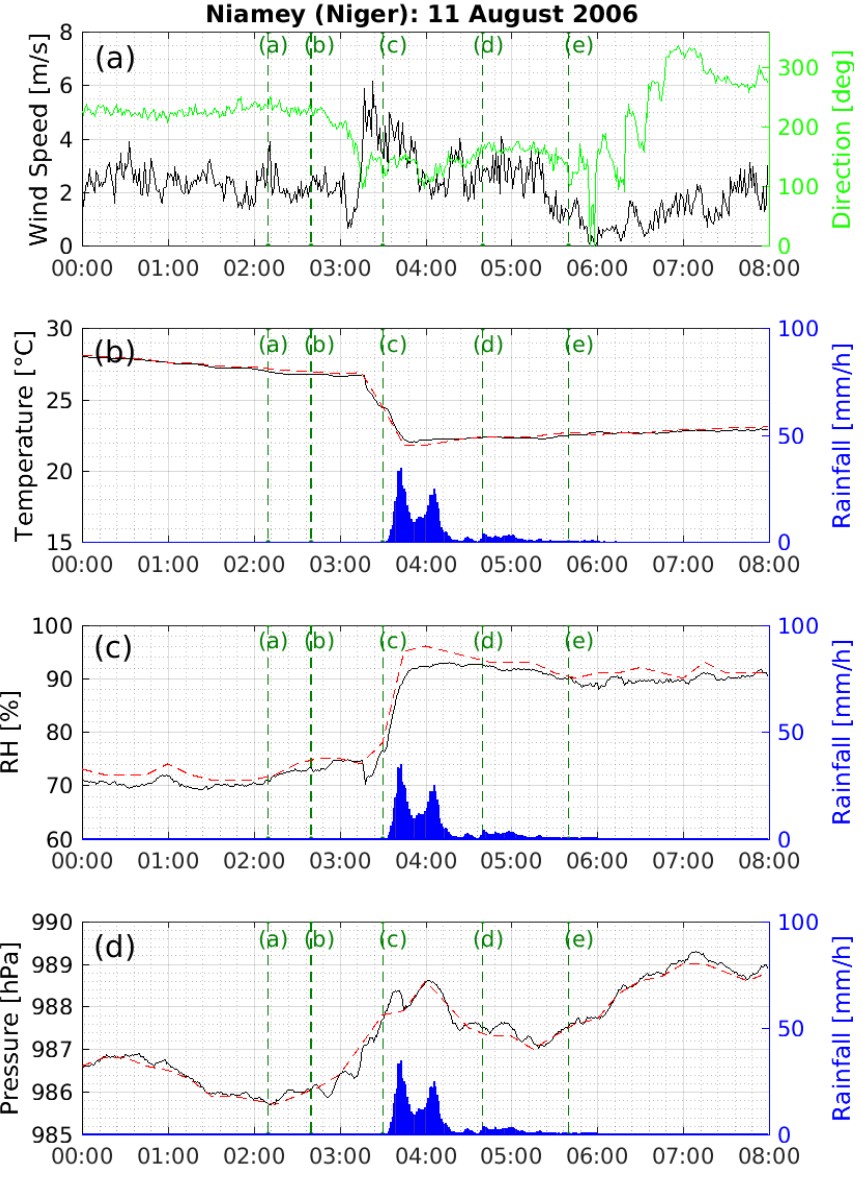

**Figure 7: Meteorological data during MCS event of 11 August 2016 at Niamey (Niger) retrieved from ARM Mobile Facility at 1-min sampling (black or green lines) and from PTU200 sensor at 15-min sampling (red dashed lines): (a) wind speed [m/s] and direction [deg.], (b) temperature [°C], (c) relative humidity [%], (d) pressure [hPa] and rainfall [mm/h] (blue bar plots). The times of the reflectivity maps given Fig. 6 are spotted by vertical dashed green lines and correspond to (a) 2:10 UTC, (b) 2:40 UTC, (c) 3:30 UTC, (d) 4:40 UTC and (e) 05:40 UTC.**





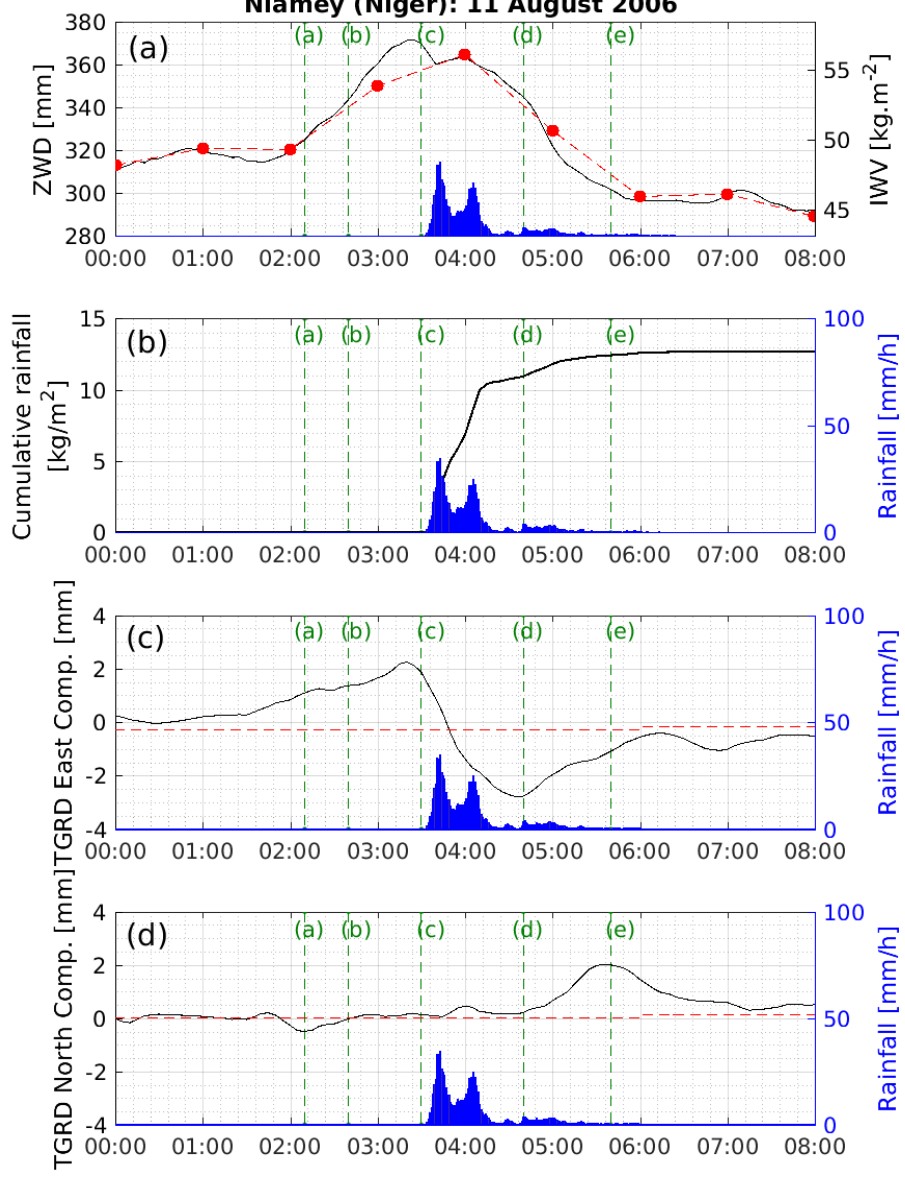

**Figure 8: Tropospheric estimates from GIPSY-OASIS (black solid line) and GAMIT (red dashed line) software during MCS event of 11 August 2016 at Niamey, Niger: (a) Zenithal Wet Delay [mm] and it equivalent IWV [kg.m⁻²] (red dots), (c) East and (d) North components of tropospheric gradients [mm]. (b) Cumulative rainfall [kg.m⁻²] (black line) is computed from the 1-min. retrieved rainfall [mm/h] (blue bar plots) of the ARM Mobile Facility. Rainfall [mm/h] (blue bar plots) in (a) is repeated from the graphs (b, c & d) but without dedicated graduation axis. The times of the reflectivity maps given Fig. 6 are spotted by vertical dashed green lines and correspond to (a) 2:10 UTC, (b) 2:40 UTC, (c) 3:30 UTC, (d) 4:40 UTC and (e) 05:40 UTC.**



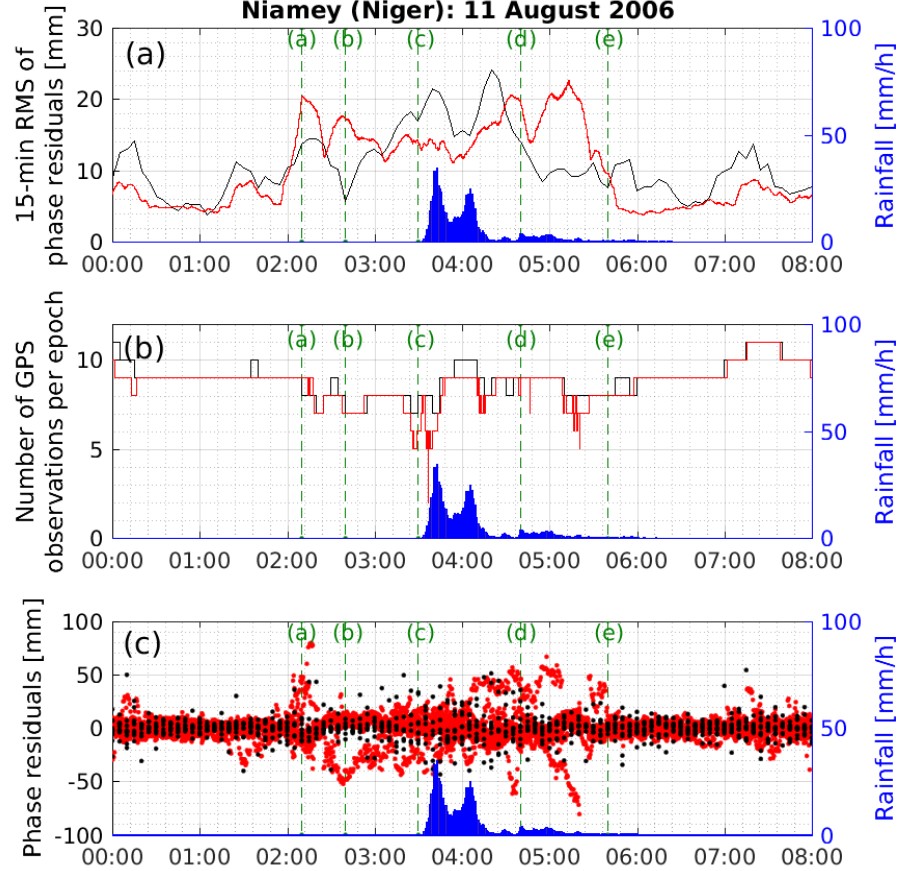

**Figure 9: Outputs from GIPSY-OASIS (black) and GAMIT (red) software during MCS event of 11 August 2016 at Niamey (Niger): (a) 15-min RMS of GPS phase residuals [mm], (b) number of GPS observations per epoch and (c) GPS phase residuals [mm] at full temporal resolution (5 min for GIPSY-OASIS and 30 sec for GAMIT). Rainfall [mm/h] (blue bar plots) is retrieved from ARM Mobile Facility. The times of the reflectivity maps given Fig. 6 are spotted by vertical dashed green lines and correspond to (a) 2:10 UTC, (b) 2:40 UTC, (c) 3:30 UTC, (d) 4:40 UTC and (e) 05:40 UTC.**







1    **Figure 10: Similar to Figure 6 but with GPS post-fit phase residuals from the GAMIT processing, with 30 s sampling,**
2    **within 5-min intervals centered on the times of the radar plots.**



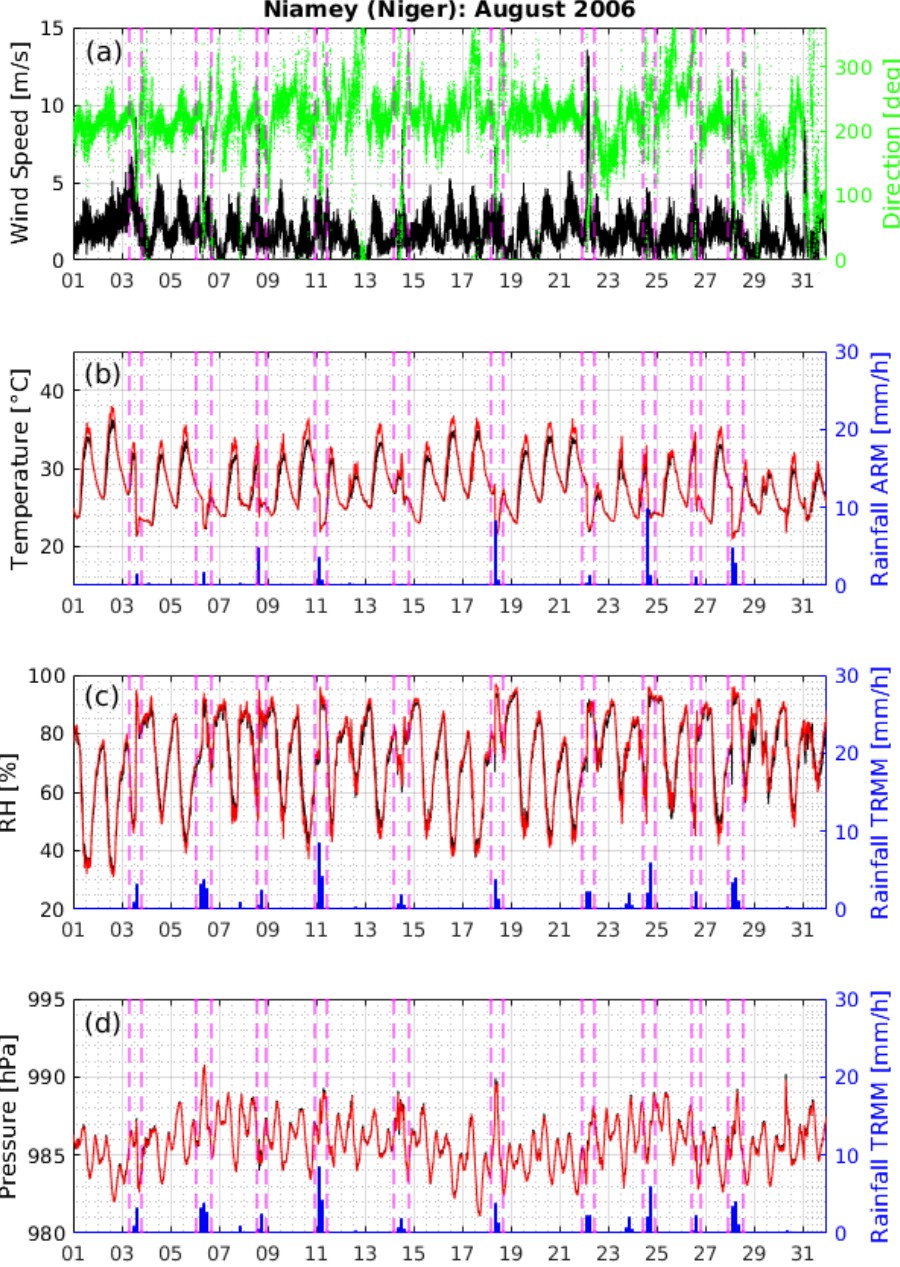

**Figure 11: Surface meteorological data from the ARM Mobile Facility with 1-min sampling (black or green lines) and from PTU200 sensor with 15-min sampling (red lines), at Niamey during August 2006: (a) Wind speed [m/s] and direction [deg.], (b) Temperature [°C], (c) Relative Humidity [%] and (d) Pressure [hPa]. 3-h rainfall [mm/h] (blue bar plots) is retrieved (b) from ARM Mobile Facility and (c, d) from the TRMM Multisatellite Precipitation Analysis. Each detected MCSs is spotted by two dashed vertical dashed magenta lines indicating the start and end of the rainfall period.**





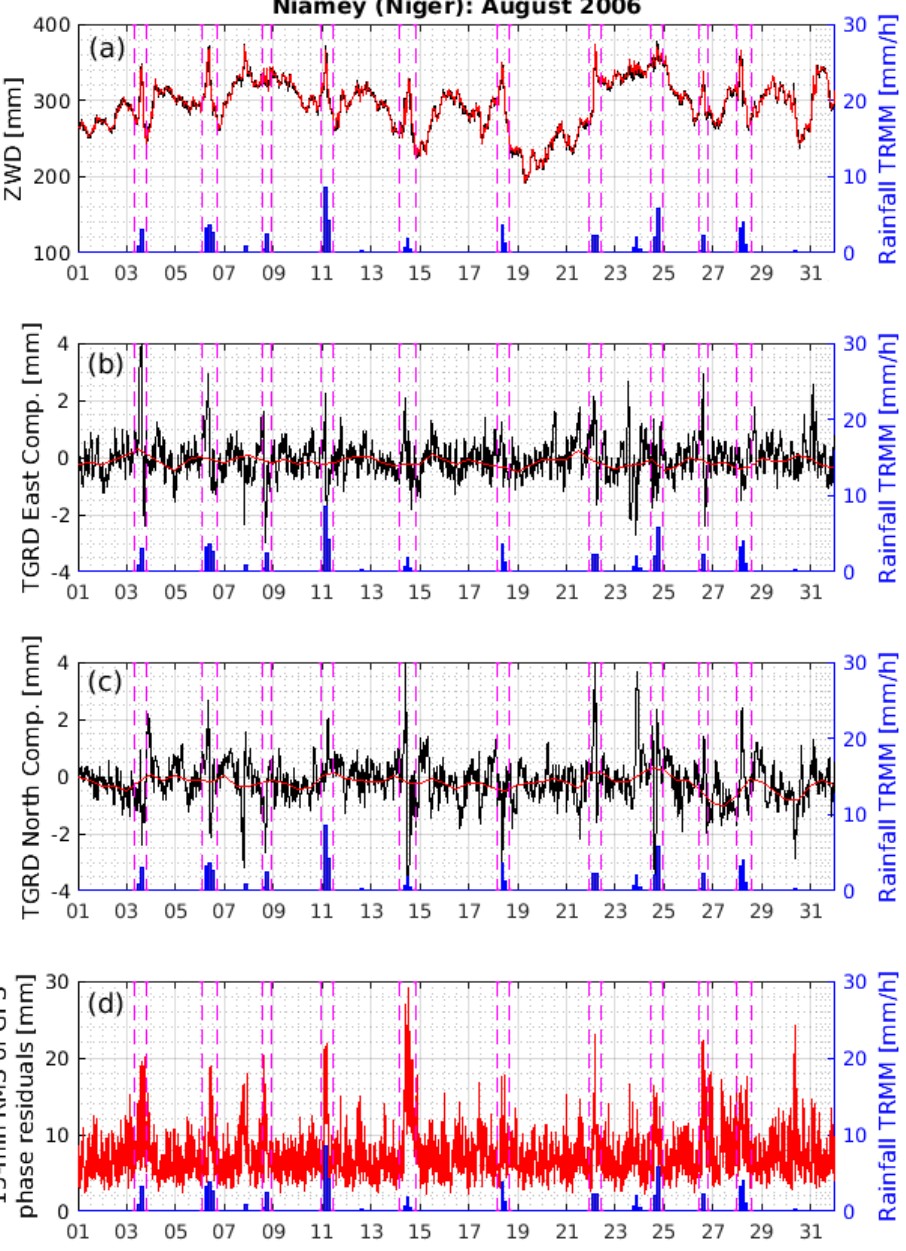

**Figure 12: Tropospheric estimates from GIPSY-OASIS (black) and GAMIT (red) at Niamey during August 2006: (a) Zenithal Wet Delay [mm], (b) East and (c) North components of tropospheric gradients [mm] and (d) 15-min RMS of GPS phase residuals [mm]. Each detected MCS is spotted by two dashed vertical magenta lines indicating the start and end of the rainfall period.**





1     **Figure 13: Composites of surface air (a) temperature [°C], (b) relative humidity [%], (c) pressure [hPa] from in-situ**
2     **PTU200 sensor, and (d) ZWD [mm], (e) East and (f) North components of the tropospheric delay gradient [mm] at all**
3     **six AMMA stations, on a 10-hour window centered on cold pool crossing time (CPCT).**





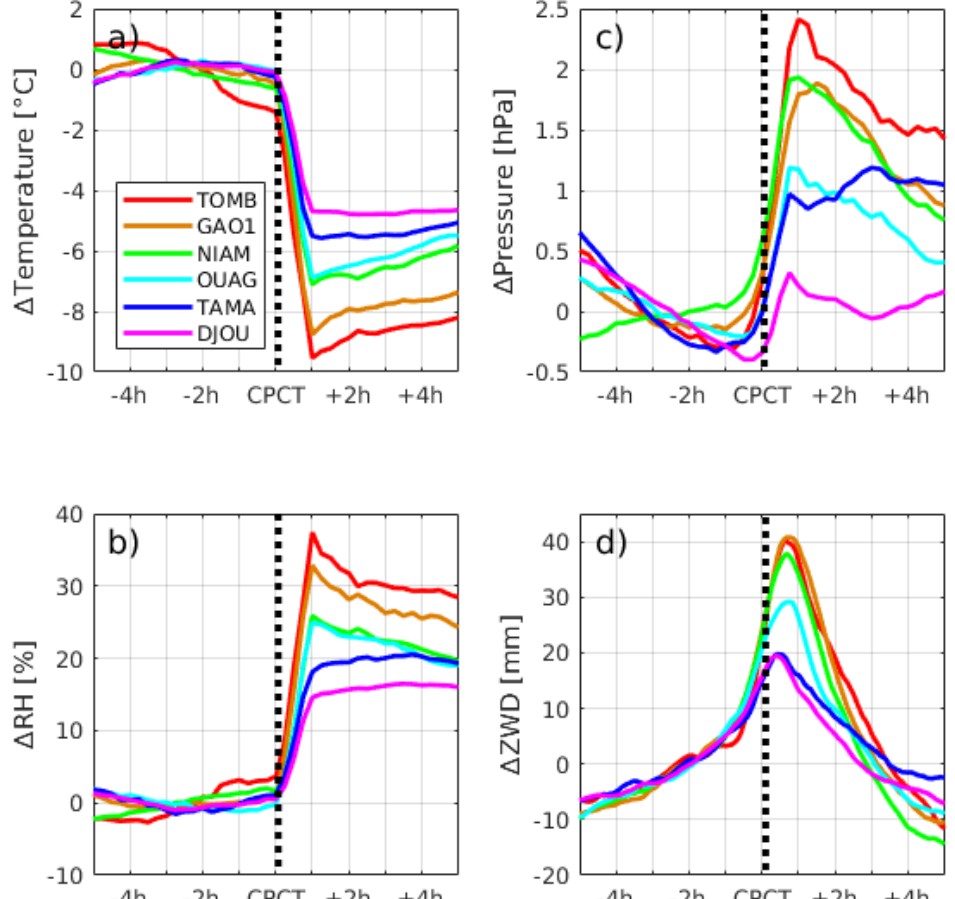

**Figure 14: Same as Fig. 13 but variables are differenced with respect to the average values computed over the 5 hours preceding the cold pool crossing time (CPCT). The average values are given in Table 3.**