# Peer review of "Sensitivity of GPS tropospheric estimates to mesoscale convective systems in West Africa"

_Atmospheric Chemistry and Physics, 2018_

## Referee Comment (RC1) · Anonymous Referee #1 · 28 Mar 2019

General comments In this paper the characteristics of GPS tropospheric estimates (ZWD and gradients) and post-fit phase residuals during the wet season of the WAM have been investigate using two different GPS approaches (regional network of GPS stations, observations in PPP mode). MCSs passages are analysed based on a case study from the AMMA period in 2006 and on a statistical approach. The aims of the investigation are clearly given. The paper is clearly structured and well written and the added value of GPS information concerning MCS analysis is quite obvious. I recommend publication of the paper - some minor recommended changes are given below.

specific comments Section 2.3.1: I am not very familiar with the two parameters tropospheric gradients and post-fit phase residuals. E.g. a gradient normally is defined by dy/dx (unit1/unit2). However, here gradients are given in mm (e.g. Figure 4). I.e. the

[Figure]

"gradients" are more related to spatial (north-south, east-west) inhomogeneities rather than real gradients. Therefore, I would recommend to add some more information how to interpret the "gradient" data (can any information be given over which horizontal distance the values occur?). Concerning post-fit phase residuals: some more information what it really means and how to interpret the values would help readers which are not that familiar with this kind of GPS data analysis. Page 13, line 9: I wonder why you discuss ZWD together with relative humidity and not with absolute or specific humidity. As in most cases, i.e. in the WAM region, too, the IWV should be mainly determined by the humidity in the boundary-layer (this is e.g. obvious from specific humidity profiles, Schwendike et al., 2010). Thus, ZWD should show a better correlation with the near-surface absolute humidity than relative humidity does. Schwendike, J.; Kalthoff, N.; Kohler, M., 2010: The impact of mesoscale convective systems on the surface and boundary-layer structure in West Africa: Case-studies from the AMMA campaign 2006. doi:10.1002/qj.599 . See also comment beow. Figure 6 and 10 include ground clutter. Could ground clutter be removed so that backscatter from rain remains.

Technical corrections There are several typos etc. a few (not all) are listed below Page 1, line 14: should be '. . . the case of an MCS' Page 4, line 37: should be ". . . . Whose parameter is ten times .." Page 7, PTU200 data and figure 7 and page 8 lines 14-20: as the PTU200 data are not really discussed, I would recommend to remove them from the diagram. The good agreement between ARM and PTU200 data could be mentioned in one sentence (when used in sect. 4). Page 7, line 29: it would be sufficient to give wind speed with one digit "5.8 m/s" instead of "5.81 m/s" as done before. Page 8, line 2: 3:33 UTC until 6:41 UTC would be 188 minutes. Where does 182 minutes come from? Page 8, line 5: the start of the convective phase is given by 3:33 UTC. Here you give 3:32 UTC. Shouldn't the times be the same? Page 8, line 36: do you really mean 37 min? from line 3 and 4 it should be 41 min (29 min +12 min). Page 9, line 10: should read ". . . . to reach a maximum of " Page 10, line: should read " . . . . make it easy to . . .." -, line 13: delete "to" after UTC. Page 11, line 9: delete "at" ". . . . Cold pools during . . .." Figure 7: I would even here show the accumulated precip (instead of

showing three times the same precip data in 7b,c,d) because it is discussed in the text on page 8.

---

## Referee Comment (RC2) · Anonymous Referee #2 · 28 Mar 2019

**Review of *Sensitivity of GPS tropospheric estimates to mesoscale convective systems in West Africa*, by Nahmani, Bock and Guichard. ACP-2018/1242**

**General comments**

This paper assesses the variation of GPS tropospheric estimates (ZWD, ZTD gradients and IWV) in connection with passages of meso scale convective systems (MCS) above and past a set of GPS sites in West Africa. It does so for two types of GPS data processing, network versus PPP.

It is demonstrated convincingly that indeed the GPS derived tropospheric estimates are sensitive to MCSs.

It is a concise, nicely written paper that I enjoyed reading. It can be published with only minor changes.

**Specific comments**

p 3: ECMWF pressures are used to determine ZHD. Why that when every site is equipped with meteorological sensors including a barometer?

p 5 top in connection with table 1: Consider using mm throughout. At first I got fooled not noticing cm being used for the constriant on the ZWD rate of variation in table 1.

p 7 bottom: .. the relative humidity -> ... the relative surface humidity
.. humidity increases again to reach a maximum of 76 % before the first rainfall. -> .. humidity starts rising again before the first rainfall, passing 76 % before it occurs.

p 8 around line 20: I was surprised not to see wind included in "best way to identify CPCT", but much later in the manuscript realised that possibly you don't have wind observations from all the sites. If so it would be good to give that as a reason wind is not included in you id CPCT scheme.

p 8 around line 35: It is preferable to talk about convergence of air, not separate convergence of moisture. Low level convergence brings air to the column including addtional humidity, and initiate a lift, which leads to cooling, saturation, latent heat release, etc.

p 11: Possibly observations of precipitation and lightning from ground, as well as of clouds and lightning from geostationary satellites can be added to the MCS detection arsenal?!

p 12: Are there some sunrise/sunset (which can also be associated with rapid warming/cooling) limitations to this method?

p 14 line 3: There is certainly a strong change in moisture levels associated with the passage of the MCS. But which part of that is due to convergence taking place in the neighborhood of the GPS site, and which part is just due to advection of an MCS that already contains large variations in humidty past the GPS site is less clear.

p 15, end of conclusion: Your study underlines the ability of PPP to provide high frequency estimates of ZWD and ZTD gradients, which is valuable for NWP (both for verification and assimilation) and meteorologists. Consider adding a few words on this in the discussion or the conclusion.

Figure 8 a: Is the variation of the Tm used when converting from ZWD to IWV really so little that the GAMIT curve for ZWD and the points for IWV can be placed precisely on top of one another?

---

## Author Comment (AC1) · 13 Jun 2019

**Answers to Referee n°1**

The referee's comments are repeated below in black italics and our answers are given in blue.Reviewer 1 - Specific Comments XX are abbreviated as RW1-SC-XX.

**Reviewer 1 General comments**

*In this paper the characteristics of GPS tropospheric estimates (ZWD and gradients) and post-fit phase residuals during the wet season of the WAM have been investigate using two different GPS approaches (regional network of GPS stations, observations in PPP mode). MCSs passages are analysed based on a case study from the AMMA period in 2006 and on a statistical approach. The aims of the investigation are clearly given. The paper is clearly structured and well written and the added value of GPS information concerning MCS analysis is quite obvious. I recommend publication of the paper - some minor recommended changes are given below.*

We thank the referee for his positive appreciation, valuable comments and constructive suggestions that, we think, contributed to improve the manuscript. We are pleased to answer to all his comments.

**RW1-SC-01: interpretation of tropospheric gradients**

*Specific comments Section 2.3.1: I am not very familiar with the two parameters tropospheric gradients and post-fit phase residuals. E.g. a gradient normally is defined by dy/dx (unit1/unit2). However, here gradients are given in mm (e.g. Figure 4). I.e. the "gradients" are more related to spatial (north-south, east-west) inhomogeneities rather than real gradients. Therefore, I would recommend to add some more information how to interpret the "gradient" data (can any information be given over which horizontal distance the values occur?).*

**Reply to the RW1-SC-01**

To answer your comment about the units in short: the gradient parameters are in units of m because they model the effect of horizontal gradients of the refractive index on the total optical path delay along the ray between the satellite and the receiver. Path delay is in units of m. For more details about for formulation and physics see Chen and Herring, 1997.

➔ We added the following sentences and references in the Introduction where the readers not familiar with the concepts of tropospheric delay gradients can find the definition, equations, and physical background.

Page 3 lines 1-5: "Both software implement tropospheric delay models which include also gradient parameters. Gradient parameters allow to represent the effect of first order azimuthal asymmetry in the atmospheric refractive index (MacMillan, 1995; MacMillan and Ma, 1997; Chen and Herring, 1997). The spatial scale over which the GPS measurements are sensitive to atmospheric refractivity gradients is about 50 km."

**RW1-SC-02: more information about post-fit phase residuals**

*Concerning post-fit phase residuals: some more information what it really means and how to interpret the values would help readers which are not that familiar with this kind of GPS data analysis.*

**Reply to the RW1-SC-02**

The post-fit phase residuals are the residuals (observed minus computed phase) from the least-squares fit when the estimated parameters are used in the model (« post-fit »). This is very standard with all fitting methods, see e.g. https://en.wikipedia.org/wiki/Least_squares_adjustment

➔ We added a GPS-related reference at the end of the first sentence on section 2.3.2 page 6 line 21:

Kouba, J.: A guide to using International GPS Service (IGS) products. IGS Central Bureau, Pasadena (available at http://igscb.jpl.nasa.gov/igscb/resource/pubs/GuidetoUsingIGSProducts.pdf), 2003.

**RW1-SC-03: "ZWD should show a better correlation with the near-surface absolute humidity than relative humidity does "**

*Page 13, line 9: I wonder why you discuss ZWD together with relative humidity and not with absolute or specific humidity. As in most cases, i.e. in the WAM region, too, the IWV should be mainly determined by the humidity in the boundary-layer (this is e.g. obvious from specific humidity profiles, Schwendike et al., 2010). Thus, ZWD should show a better correlation with the near-surface absolute humidity than relative humidity does. Schwendike, J.; Kalthoff, N.; Kohler, M., 2010: The impact of mesoscale convective systems on the surface and boundary-layer structure in West Africa: Case-studies from the AMMA campaign 2006. doi:10.1002/qj.599 .*

**Reply to the RW1-SC-03**

We fully agree with the reviewer that specific humidity is more directly linked to ZWD. Note that the aim of Table 3 is more broadly to compare results at sites located in distinct climates just before the arrival of the cold pools, not so much to analyse correlations with surface air humidity. In fact, we used relative humidity following Lothon et al. (2011), as the arrival of the cold pool is also quite well captured with relative humidity (e.g. their Fig. 5 or our Figs. 7 and 11).

**RW1-SC-04 : "ground clutter"**

*Figure 6 and 10 include ground clutter. Could ground clutter be removed so that backscatter from rain remains.*

**Reply to RW1-SC-04**

You are right, a signature of clutter can be seen in the images, but we don't have the full radar data at hand, so we can't remove these features. On the other hand, it is quite clear from the successive plots that the features are stationary and cannot be confounded with the MCS. So, we think it is not necessary to change these figures.

**RW1-SC-05: Technical corrections**

*Technical corrections There are several typos etc. a few (not all) are listed below Page 1, line 14: should be '... the case of an MCS' Page 4, line 37: should be "... Whose parameter is ten times .." Page 7, PTU200 data and figure 7 and page 8 lines 14-20: as the PTU200 data are not really discussed, I would recommend to remove them from the diagram. The good agreement between ARM and PTU200 data could be mentioned in one sentence (when used in sect. 4). Page 7, line 29: it would be sufficient to give wind speed with one digit "5.8 m/s" instead of "5.81 m/s" as done before. Page 8, line 2: 3:33 UTC until 6:41 UTC would be 188 minutes. Where does 182 minutes come from? Page 8,*

*line 5: the start of the convective phase is given by 3:33 UTC. Here you give 3:32 UTC. Shouldn't the times be the same? Page 8, line 36: do you really mean 37 min? from line 3 and 4 it should be 41 min (29 min +12 min). Page 9, line 10: should read ".... to reach a maximum of " Page 10, line: should read " .... make it easy to ...." -, line 13: delete "to" after UTC. Page 11, line 9: delete "at" ".... Cold pools during ...." Figure 7: I would even here show the accumulated precip (instead of showing three times the same precip data in 7b,c,d) because it is discussed in the text on page 8.*

**Reply to RW1-SC-05**

➔ *Page 1, line 14: should be '... the case of an MCS'* / **done**
➔ *Page 4, line 37: should be "... Whose parameter is ten times .."* / **done**

➔ PTU200 data are removed from Figure 7 p37 & Figure 11 p42 and the text has been modified accordingly section 3.1 p7 line 21. To meet recommendations of the two reviewers, The last paragraph of section 3.1 page 8 line 17-25 becomes:

"There is really added value of having the high sampling data from ARM-MF to capture details of the internal dynamics of the MCS. However, such data are only available at Niamey (Niger) for 2006 only. The data retrieved from the PTU200 sensor are in good agreement with data from ARM-MF but the 15-min sampling is not sufficient to detect that the surface temperature drops in two consecutive stages or the sudden and brief drop in relative humidity (not shown). Moreover, because of GPS stations were aimed at providing IWV data, the wind has not been recorded. The best way to identify the CPCT with only PTU200 data at the other AMMA GPS stations is thus to detect significant drops in surface temperature over a period between 30 minutes to 1 hour followed by strong rainfall (see section 4)."

➔ *Page 7, line 29: wind speed with one digit "5.8 m/s" instead of "5.81 m/s"* / **done**
➔ *Page 8, line 2: 3:33 UTC until 6:41 UTC would be 188 minutes. Where does 182 minutes come from?* You are right, logically, the sum should have matched but 6:41 UTC is the last time where rainfall more and more scattered is observed. The difference of 6 minutes corresponds to a period with no rainfall between 3:33 UTC and 6:41 UTC. ->For a better comprehension, the sentence becomes: "From 03:33 UTC until 06:41 UTC, the MCS events characterized by a two-phase rainfall pattern produces successively by its convective and stratiform parts."

➔ Page 8, line 5: the start of the convective phase is given by 3:33 UTC. Here you give 3:32 UTC. Shouldn't the times be the same? /You are right. The correction is made now.

---

## Author Comment (AC2) · 13 Jun 2019

**Answers to Referee n°2**

The referee's comments are repeated below in black italics and our answers are given in blue.Reviewer 1 - Specific Comments XX are abbreviated as RW1-SC-XX.

**Reviewer 2 General comments**

*This paper assesses the variation of GPS tropospheric estimates (ZWD, ZTD gradients and IWV) in connection with passages of mesoscale convective systems (MCS) above and past a set of GPS sites in West Africa. It does so for two types of GPS data processing, network versus PPP.*
*It is demonstrated convincingly that indeed the GPS derived tropospheric estimates are sensitive to MCSs.*
*It is a concise, nicely written paper that I enjoyed reading. It can be published with only minor changes.*

We thank the referee for his positive appreciation and all comments which helpedus to improve the manuscript. Please find below detailed clarifications and responses:

**RW2-SC-01: "why ECMWF pressure?"**

*p 3: ECMWF pressures are used to determine ZHD. Why that when every site is equipped with meteorological sensors including a barometer?*

**Reply to RW2-SC-01**

You are right, the surface pressure observations could actually be used to compute the a priori ZHD values but this is not done for two reasons. First, it is a standard and convenient procedure to use ECMWF ZHD estimates as a priori and estimate ZWD during the processing because all GNSS software can download and handle automatically the ECMWF data provided by Tech. Univ. Vienna, (simultaneously with the VMF1 mapping function parameters). Second, for the six stations analysed in this study, the ECMWF surface pressure field is very accurate, the comparison to our observations is at the level -0.9 ± 1.1 hPa, and the ECMWF data have no gaps contrary to the observations.

**RW2-SC-02: "cm -> mm"**

*p 5 top in connection with table 1: Consider using mm throughout. At first I got fooled not noticing cm being used for the constraint on the ZWD rate of variation in table 1.*

**Reply to RW2-SC-02: done**

**RW2-SC-03: "relative surface humidity"**

*p 7 bottom: .. the relative humidity -> ... the relative surface humidity .. humidity increases again to reach a miximum of 76 % before the first rainfall. -> .. humidity starts rising again before the first rainfall, passing 76 % before it occurs.*

**Reply to RW2-SC-03: done**

**RW2-SC-04: why no wind data?**

*p 8 around line 20: I was surprised not to see wind included in "best way to identify CPCT", but much later in the manuscript realised that possibly you don't have wind observations from all the sites. If so it would be good to give that as a reason wind is not included in you id CPCT scheme.*

➔ **To meet recommendations of the two reviewers**, The last paragraph of section 3.1 page 8 line 17-25 becomes:

There is really added value of having the high sampling data from ARM-MF to capture details of the internal dynamics of the MCS. However, such data are only available at Niamey (Niger) for 2006 only. The data retrieved from the PTU200 sensor are in good agreement with data from ARM-MF but the 15-min sampling is not sufficient to detect that the surface temperature drops in two consecutive stages or the sudden and brief drop in relative humidity (not shown). Moreover, because of GPS stations were aimed at providing IWV data, the wind has not been recorded. The best way to identify the CPCT with only PTU200 data at the other AMMA GPS stations is thus to detect significant drops in surface temperature over a period between 30 minutes to 1 hour followed by strong rainfall (see section 4).

**RW2-SC-05: talk about convergence of air, not separate convergence of moisture**

*p 8 around line 35: It is preferable to talk about convergence of air, not separate convergence of moisture. Low level convergence brings air to the column including additional humidity, and initiate a lift, which leads to cooling, saturation, latent heat release, etc.*

**Reply to RW2-SC-05**

The sentence was changed to: "The dynamics of the active convective cell involves a low-level convergence of moist air from its immediate environment, which initiate a lift and leads to the significant rainfall amount of 37 mm."

**RW2-SC-06: more data to detect MCS**

*p 11: Possibly observations of precipitation and lightning from ground, as well as of clouds and lightning from geostationary satellites can be added to the MCS detection arsenal?!*

**Reply to RW2-SC-06**

You are right. Such observations could be added to strengthen the MCS detection. Thefollowing sentence wasadded at the end of section 4.1: « We note that precipitation and lightning from ground, as well as of clouds and lightning from geostationary satellites could be added to the MCS detection arsenal. These additional information will be considered in the future.»

**RW2-SC-07: limitations of the CPCT detection**

*p 12: Are there some sunrise/sunset (which can also be associated with rapid warming/cooling) limitations to this method?*

**Reply to RW2-SC-07**

RW2 is quite right to question out the possible limitations of the detection of MCS and CPCT, especially due to the sub-daily variations of soil surface temperature due to the sun (sunrise and sunset). For this reason, the detection method is done in two steps:

➔ The first step is the detection of sufficiently important precipitations which implies the presence of a significant cloud cover attenuating the variations of the soil surface temperature due to the sun.

➔ The second step is the search for a significant drop in soil surface temperature.

The detection method would not have been valid if it had relied solely on variations in soil temperature.

We add in the manuscript, section 4.1, p11, 2$^{nd}$ paragraph:

"The first detection step is fundamental because an important rainfall implies the presence of a significant cloud cover which attenuates the variations of the surface temperature due solar radiation and thus avoids the false detections and fails due to the daily solar cycle."

**RW2-SC-08: convergence/advection ?**

*p 14 line 3: There is certainly a strong change in moisture levels associated with the passage of the MCS. But which part of that is due to convergence taking place in the neighborhood of the GPS site, and which part is just due to advection of an MCS that already contains large variations in humidity past the GPS site is less clear.*

**Reply to RW2-SC-08**

This is a valuable question. From those time series alone, it is not possible to conclude. However, we can note from Fig. 14 that the fluctuations of ZWD occurs on sub-daily scales (typically about 6-12h), which suggest that the associated spatial scale are sub-synoptic.

➔ Initial version: Though there is a strong moisture convergence associated with the passage of the MCS, the tendency after it is a slightly drier air column, especially in the more arid climate (this is consistent with typical signatures found in sounding data, not shown). The ZWD peak is also narrower from the southern to the northern sites, which suggests faster MCS in the north (again consistent with existing studies, e.g. (Maranan et al., 2018)).

➔ Modified version:Though there is a strong moisture convergence associated with the passage of the MCS, the tendency after it is a slightly drier air column, especially in the more arid climate (this is consistent with typical signatures found in sounding data, not shown). The ZWD peak is also **relatively narrow (less than 12h), and** narrower from the southern to the northern sites, which suggests faster MCS in the north (again consistent with existing studies, e.g. (Maranan et al., 2018)). **It also implies that the moisture convergence occurs at sub-synoptic scales.**

**RW2-SC-09:**

*p 15, end of conclusion: Your study underlines the ability of PPP to provide high frequency estimates of ZWD and ZTD gradients, which is valuable for NWP (both for verification and assimilation) and meteorologists. Consider adding a few words on this in the discussion or the conclusion.*

**Reply to R2-SC-10**

The benefit of high frequency was already mentioned in the conclusion (P15L8-9): "Thanks to the high temporal sampling of the gradient estimates, the GIPSY-OASIS solution provides additional information on the atmospheric anisotropy."

**As requested we reformulated the last paragraph:**

"To conclude, this study showed that the high frequency estimates of ZWD and tropospheric delay gradients are relevant for climate monitoring and documentation of intense weather events such as

MCSs. They could also be used to study other rapid meteorological processes and for the verification and assimilation in numerical weather prediction models."

**RW2-SC-10: ZWD -> IWV which Tm?**

*Figure 8 a: Is the variation of the Tm used when converting from ZWD to IWV really so little that the GAMIT curve for ZWD and the points for IWV can be placed precisely on top of one another?*

**Reply to R2-SC-10**

In this figure we used a constant value for the conversion coefficient. The goal is to give an idea of the amount IWV the ZWD variations represent. This is actually the only place where we show IWV values.